## RESEARCH ARTICLE

# A C-terminal cytoplasmic retention motif and nuclear localization signal regulates nuclear import of TP53INP2

Birendra Kumar Shrestha, Eva Sjøttem, Hallvard Lauritz Olsvik, Isaac Odonkor, Aud Øvervatn, Hanne Britt Brenne, Jack-Ansgar Bruun, Trond Lamark and Terje Johansen*

## ABSTRACT

Tumor protein p53 inducible nuclear protein 2 (TP53INP2; also known as DOR) is a multifunctional protein involved in transcriptional coactivation, ribosomal RNA synthesis and autophagy, regulated by subcellular localization. Using CRISPR/Cas9-generated TP53INP2-knockout HeLa cells reconstituted with EGFP–TP53INP2, we show that TP53INP2 is predominantly degraded by nuclear proteasomes under basal conditions. Under stress, including starvation and various chemical stress inducers, TP53INP2 accumulates in the cytoplasm independently of ATG5, CRM1-mediated export, phosphorylation, ubiquitylation or acetylation. We identify a nuclear localization signal (NLS) overlapping a nucleolar localization signal (NoLS) in the C-terminus, which mediates nuclear import and nucleolar enrichment. Deletion of this region redirects TP53INP2 to LC3B-positive puncta. A conserved nine-amino-acid cytoplasmic retention motif (CRM) in the C-terminus prevents nuclear re-entry under stress. This motif and regulation of subcellular localization is conserved in the related TP53INP1 protein. Fluorescence recovery after photobleaching (FRAP) and importin-binding assays show that nutrient starvation disrupts nuclear import of TP53INP2. Finally, we show that starvation enhances TP53INP2 translation via the m6A demethylase FTO, without altering mRNA stability. These findings uncover coordinated regulation of TP53INP2 localization and turnover by cellular stress.

KEY WORDS: Autophagy, Cytoplasmic retention, DOR, LC3B, Nuclear localization, Proteasome, TP53INP2, TP53INP1

## INTRODUCTION

Tumor protein p53 inducible nuclear protein 2 (TP53INP2), also called diabetes and obesity-related (DOR), was originally identified as a nuclear protein expressed in tissues with a high metabolism level. It acts as a transcription coactivator of the thyroid hormone receptor and regulates thyroid hormone function (Baumgartner et al., 2007; Francis et al., 2010). In a transgenic mouse model, muscle-specific overexpression of TP53INP2 leads to reduced muscle mass and deletion of it leads to muscle hypertrophy (Sala

Autophagy Research Group, Department of Medical Biology, UiT The Arctic University of Norway, 9037 Tromsø, Norway.

*Author for correspondence (terje.johansen@uit.no)

B.K.S., 0000-0002-8972-0308; E.S., 0000-0003-2668-1708; H.L.O., 0000-0003-3489-7461; I.O., 0009-0008-5334-1585; J.-A.B., 0000-0003-0614-2790; T.L., 0000-0001-6338-3342; T.J., 0000-0003-1451-9578

et al., 2014). TP53INP2 is also linked to adipose cell differentiation. It acts as a negative regulator of adipogenesis by promoting sequestration of GSK3β in an endosomal sorting complexes required for transport (ESCRT)-dependent pathway (Romero et al., 2018). TP53INP2 sensitizes cells to apoptosis induced by death receptor ligands (Ivanova et al., 2019). Nuclear TP53INP2 is enriched in nucleoli (Mauvezin et al., 2012), where it facilitates formation of the RNA polymerase I preinitiation complex on rDNA promoters to promote ribosome biogenesis when mammalian target of rapamycin (mTOR) is active (Xu et al., 2016).

Upon mTOR inhibition by starvation or treatment with the inhibitor rapamycin, TP53INP2 is reported to interact with ATG8 family proteins and VMP1 to promote formation of autophagosomes (Nowak et al., 2009). The complete redistribution of TP53INP2 from the nucleus to the cytoplasm upon mTOR inhibition has raised the question of whether TP53INP2 has dual roles in cell anabolism and catabolism determined by its subcellular localization (Mauvezin et al., 2010; Xu and Wan, 2020). TP53INP2 is related to TP53INP1 (30% amino acid sequence identity) with *TP53INP2* being present in metazoans and *TP53INP1* arising from a gene duplication in the common ancestor of vertebrates (Sancho et al., 2012). TP53INP1 is a stress-induced tumor suppressor that regulates p53 and p73 (also known as TP53 and TP73, respectively) transcriptional activity (Saadi et al., 2015). In the nucleus, it promotes HIPK2-mediated Ser46 phosphorylation of p53, enhancing p53 stability and driving cell cycle arrest and apoptosis (Shahbazi et al., 2013). Its expression is often downregulated in cancers, highlighting its role in stress responses and tumor suppression (Shahbazi et al., 2013). One of two highly conserved regions in these two proteins (amino acid residues 28–42 in TP53INP2) harbors an LC3-interacting region (LIR) motif which binds strongly to the human ATG8 family proteins (Sancho et al., 2012). Upon transient overexpression in HeLa cells both TP53INP1 and 2 colocalize with LC3 proteins in the cytoplasm when autophagy is induced by nutrient starvation or mTOR inhibition with rapamycin. Neither wild-type nor the LIR mutant TP53INP2 is degraded by autophagy (Sancho et al., 2012); rather TP53INP2 has a short half-life of ~4 h and is degraded by the proteasome (Mauvezin et al., 2010).

LC3B (also known as MAP1LC3B) is reported to shuttle between the nucleus and the cytoplasm (Drake et al., 2010; Huang et al., 2015). Upon nutrient deprivation, nuclear LC3B becomes deacetylated at K49 and K51 by Sirt1. This is reported to allow LC3B to bind to nuclear TP53INP2, which then mediates transport of LC3B out of the nucleus (Huang et al., 2015). Recently it has been reported that cytoplasmically localized TP53INP2 is recruited to early autophagic membranes by LC3B, where it promotes the interaction between LC3B and ATG7 thereby facilitating autophagosome formation (You et al., 2019).

In this study, we mapped the amino acid residues directing nuclear (NLS) and nucleolar (NoLS) localization of TP53INP2. We

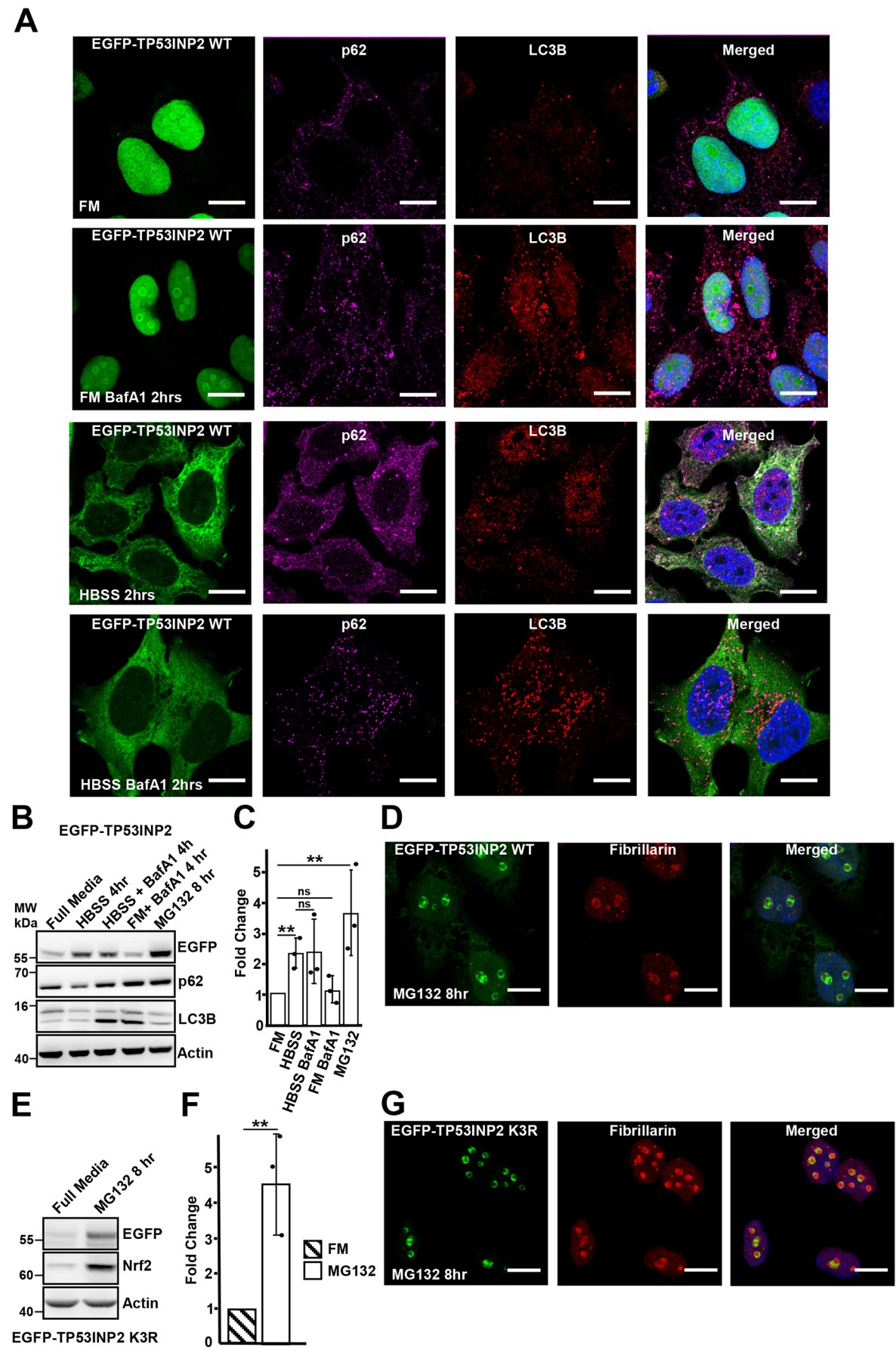

Fig. 1. See next page for legend.

**Fig. 1. TP53INP2 is degraded by the proteasome in the nucleolus but accumulates in the cytoplasm upon inhibition of mTOR.** (A) TP53INP2 does not colocalize with p62 or LC3B puncta upon starvation. TP53INP2 KO cells reconstituted with EGFP–TP53INP2 were treated with 1 µg/ml doxycycline for 24 h to induce expression. Cells were kept in full medium (FM) or treated as indicated, stained with p62 and LC3B antibodies, and analyzed by confocal microscopy. BafA1 was used at 200 µM for 2 h in HBSS-treated cells where indicated. (B) TP53INP2 is mainly degraded by the proteasome but is strongly stabilized upon starvation. After induction of EGFP–TP53INP2 with doxycycline, cells were treated as indicated and cell extracts analyzed by western blotting using the indicated antibodies. (C) Quantification of immunoblot (*n*=3 replicates) from B using ImageJ. The bars represent the mean±s.d. of band intensities relative to the actin loading controls. (D) EGFP–TP53INP2 accumulates in the nucleolus upon proteasomal inhibition. After induction of EGFP–TP52INP2 with doxycycline, cells were treated with 10 µM MG132 for 8 h, stained with fibrillarin antibodies, and analyzed by confocal imaging. (E) Mutation of all three lysine residues, K165R K187R and K204R (K3R), in TP53INP2 does not prevent its degradation by the proteasome. Cells expressing EGFP–TP53INP2 K3R were induced with doxycycline, treated as indicated and analyzed by western blotting using the indicated antibodies. (F) Quantification of immunoblot analysis from three replicates. The bars represent the mean±s.d. of band intensities relative to the actin loading controls. (G) TP53INP2 accumulates in the nucleolus upon inhibition of proteosomal degradation by MG132 treatment for 8 h. HeLa FlpIn EGFP TP53INP2 K3R cells were treated with 10 µM MG132 for 8 h, stained with antibody for the nucleolar protein fibrillarin and analyzed by confocal microscopy. Images in A, D and G representative of three repeats. Scale bars: 10 µm. \*\**P*<0.005, \**P*<0.01; ns, not significant (one-way ANOVA followed by Tukey's multiple comparison test).

established TP53INP2-knockout (KO) cells reconstituted with inducible expression at near endogenous level of various EGFP–TP53INP2 constructs. To our surprise, we found that TP53INP2 is not exported from the nucleus upon nutrient starvation. In contrast, nuclear import of newly synthesized TP53INP2 is inhibited whereas the nuclear-localized TP53INP2 is degraded by the proteasome. Mutation of all three lysine residues to arginine did not impair the starvation-induced cytoplasmic retention. The cytoplasmic retention of TP53INP2 was only impaired by deletion of the C terminal 9 amino acids. This C-terminal cytoplasmic retention motif (CRM) is also conserved in TP53INP1 and regulates its nucleocytoplasmic shutting under nutrient-deprived conditions. Besides starvation (leading to mTOR inhibition), several other cellular stressors induced cytoplasmic retention of TP53INP2 without causing mTOR inhibition. This cytoplasmic pool of TP53INP2 was not degraded by autophagy. However, overexpression of TP53INP2 in the cytoplasm leads to its recruitment to LC3B-positive puncta.

## RESULTS
### TP53INP2 is degraded by the proteasome in the nucleus and accumulates in the cytoplasm upon cellular stress
The multifaceted roles of TP53INP2, acting as a transcriptional coactivator in the cell nucleus, as a promoter of rRNA transcription in the nucleolus, and as a regulator of autophagy in the cytoplasm, seem to be regulated by subcellular localization (Mauvezin et al., 2010; Nowak et al., 2009; Xu et al., 2016). To investigate the molecular mechanisms regulating TP53INP2 localization and autophagic activity, we first established CRISPR/Cas9-mediated HeLa FlpIn KO cells of TP53INP2. KO was verified by genomic DNA sequencing as the expression level in HeLa FlpIn cells is too low to be detected by commercially available antibodies (Fig. S1A). We measured the autophagic flux in the KO cells by monitoring the levels of the autophagy receptors p62 (also known as SQSTM1) and

CALCOCO1, and the lipidation of LC3B by western blotting, in full medium (FM) and starved conditions in Hank's balanced salt solution (HBSS), which lacks serum, amino acids and glucose. However, no differences were observed in the KO cells compared to wild-type (WT) cells (Fig. S1B,C). Hence, KO of TP53INP2 does not have a detectable impact on the autophagy flux in FM or starvation in HeLa cells. Recently, TP53INP2 was shown to have a role in ribosomal biogenesis and cell proliferation (Xu et al., 2016). To confirm that the TP53INP2 KO cells have a similar phenotype, cell proliferation of the KO cells was analyzed and compared to WT cells (Fig. S1D). In line with previous reports, the TP53INP2 KO cells had a slower proliferation rate than WT cells and a decreased transcription of 47S rRNA (Fig. S1E).

Previous studies have shown that TP53INP2 redistributes to the cytoplasm upon mTOR inhibition (Mauvezin et al., 2010). To determine which factors or modifications of TP53INP2 mediate this redistribution, we reconstituted the HeLa FlpIn TRex TP53INP2 KO cells with EGFP–TP53INP2. The expression of EGFP–TP53INP2 was induced by doxycycline treatment and kept at a low level to obtain an expression level as close to the endogenous level as possible. Confocal fluorescence microscopy of the reconstituted cells in FM and upon mTOR inhibition by starvation (HBSS) or treatment with mTOR inhibitors Torin1 and PP242, revealed that redistribution of EGFP–TP53INP2 from the nucleus to the cytoplasm is regulated by mTOR activity (Fig. 1A; Fig. S2A,B). To test whether this response was specific for mTOR inhibition, we tested several other stressors. These included the uncoupling reagent and strong mitophagy inducer carbonyl cyanide m-chlorophenylhydrazone CCCP (Fig. S2C,D), the sarco-endoplasmic reticulum $Ca^{2+}$-ATPase (SERCA) pump inhibitor and ER stress inducer thapsigargin (TG) (Fig. S2E,F), sodium arsenite ($NaAsO_2$), which is a strong oxidative stress inducer widely used to stimulate stress granule (SG) formation (Fig. S2G,I), puromycin (Puro), acting as a premature translation terminator resulting in short defective ribosomal products that are ubiquitylated and sequestered into p62 bodies (Fig. S2H,I), and the natural isothiocyanate sulforaphane (SFN), which leads to stabilization and nuclear translocation of the antioxidant response transcription factor NRF2 (Fig. S2J). All these stressors, except for SFN, led to cytoplasmic redistribution of TP53INP2, although mTOR was not inhibited, as shown by blotting for S65 phosphorylation of the mTOR substrate 4EBP1 (also known as EIF4EBP1) (Fig. S2B,D,F,I).

EGFP–TP53INP2 showed a predominantly diffuse cytoplasmic staining and formed only a very few distinct puncta during starvation and upon treatment with the other stressors leading to cytoplasmic redistribution. The autophagy receptor p62 and LC3B formed puncta that increased upon starvation and upon treatment with the lysosomal inhibitor Bafilomycin A1 (BafA1) (Fig. 1A). Hence, autophagy is normal in the cell line. In contrast, EGFP–TP53INP2 overexpressed by transient transfection in the KO cells, formed puncta that colocalized with LC3B in the cytoplasm upon starvation and accumulated in response to BafA1 treatment (Fig. S3). Therefore, overexpression of TP53INP2 impairs its normal subcellular localization pattern.

To determine whether EGFP–TP53INP2 is an autophagic substrate, the level of EGFP–TP53INP2 upon starvation and BafA1 treatment was measured by western blotting. To our surprise, the expression level of EGFP–TP53INP2 was strongly increased upon starvation and unaffected by BafA1 treatment (Fig. 1B,C). In contrast, inhibition of proteasomal degradation by MG132 treatment led to increased levels of EGFP–TP53INP2

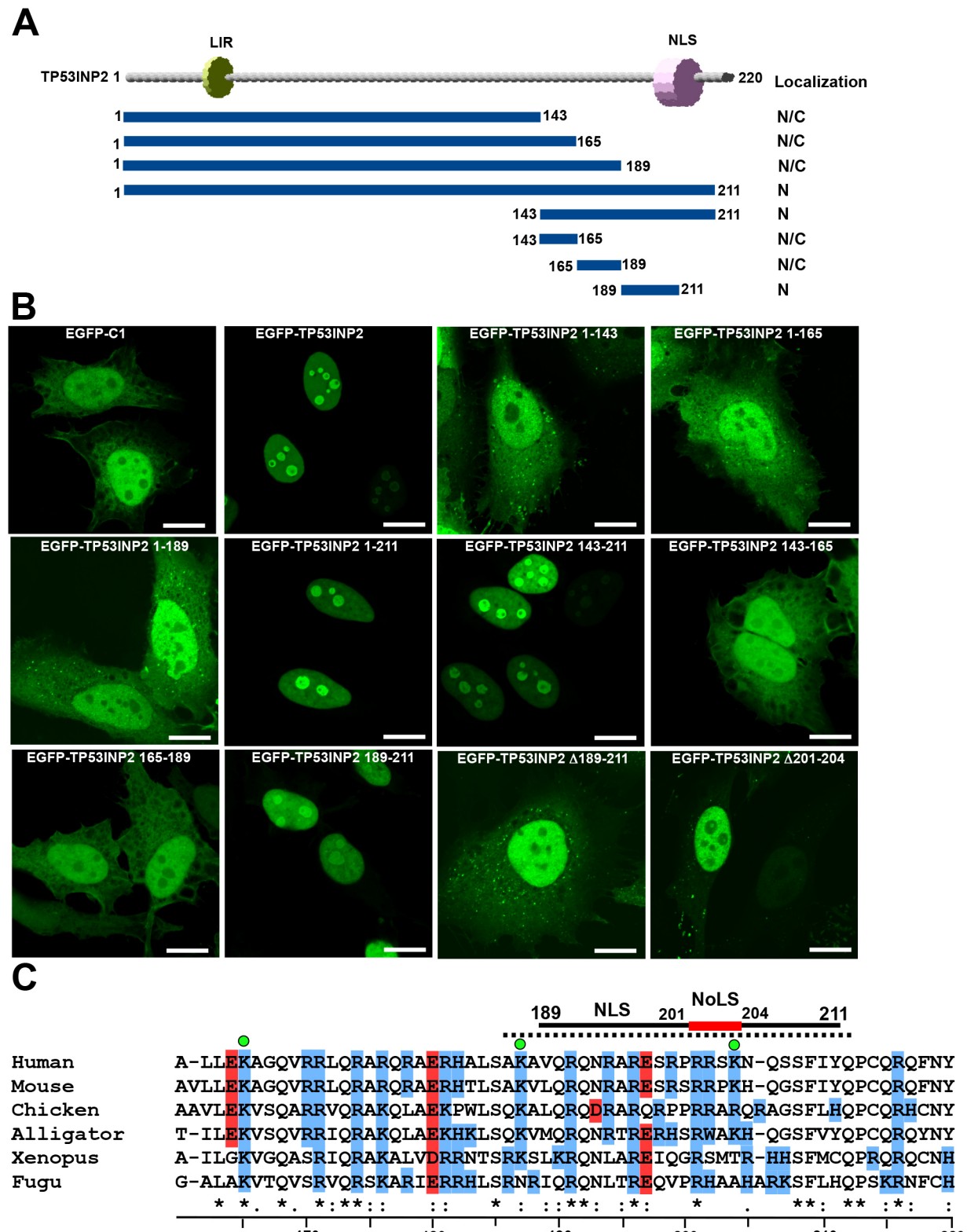

**Fig. 2.** See next page for legend.

(Fig. 1B,C). Furthermore, confocal microscopy imaging of MG132 treated cells showed accumulation of EGFP-TP53INP2 in fibrillarin-enriched structures in the nucleoli (Fig. 1D). Proteins are degraded by ubiquitin-dependent and -independent pathways. To test whether ubiquitylation directed the nucleolar degradation of TP53INP2, we mutated all three lysine (K) residues to arginine (R) residues, and reconstituted KO cells with the K3R triple mutant. Interestingly, the K3R mutant also displayed increased accumulation of TP53INP2 in nucleoli upon MG132 treatment, indicating ubiquitin-independent degradation (Fig. 1E–G). Altogether, our

**Fig. 2. Overlapping NLS and NoLS signals are located in the C-terminal region of TP53INP2.** (A) Schematic diagram of TP53INP2 showing deletion constructs used to identify the nuclear localization signal (NLS) and nucleolar localization signal (NoLS) in TP53INP2. Localization of different deletion constructs in the nucleus (N) or cytoplasm (C) of transfected cells is indicated to the right. (B) HeLa cells transiently transfected with the indicated EGFP–TP53INP2 constructs were analyzed by confocal microscopy at 24 h post transfection. Images representative of three repeats. Scale bars: 10 μm. (C) Mapping of NLS and NoLS motifs in TP53INP2. The extent of the motifs is shown above a sequence alignment indicating evolutionary conservation of the NLS and NoLS motifs in higher vertebrates. The stippled line indicates the NoLS predicted by the nucleolar localization sequence detector (Scott et al., 2011). The three lysine residues in human TP53INP2 are indicated by green dots. Basic residues are indicated in blue and acidic ones in red. The amino acid numbering below the alignment is based on the human TP53INP2 sequence. * indicates identical residues in all sequences, : indicates identical residues in almost all sequences.

results show that starvation, ER-stress, mitochondrial stress, strong oxidative stress and proteostatic stress lead to cytoplasmic accumulation of EGFP–TP53INP2 and that the nuclear pool of TP53INP2 is degraded by the proteasome.

### A nucleolar localization signal is embedded within a nuclear localization signal in the C-terminal region of TP53INP2

To facilitate the nuclear pore complex (NPC)-mediated active import of a protein requires a nuclear localization signal (NLS) that interacts with importin proteins. The first described NLS was identified in SV40T antigen (PKKKRKV) (Kalderon et al., 1984). To map the NLS directing nuclear import of TP53INP2 we made several deletions constructs of TP53INP2 fused to EGFP (Fig. 2A). These expression constructs were transfected into HeLa cells and the subcellular localization of the various constructs was analyzed by confocal fluorescence microscopy (Fig. 2B). It is important to note that the EGFP–TP53INP2 construct with a molecular mass of 50 kDa is also able to diffuse though the nuclear pores and will, without an active NLS, display a nuclear–cytoplasmic localization. The N-terminal part of TP53INP2 (amino acids 1–143, 1–165 and 1–189) displayed such a nuclear–cytoplasmic localization pattern, like EGFP alone (Fig. 2B). However, extending the expression construct to include the region 189–211 resulted in a dominant nuclear localization suggesting the presence of an NLS motif within the region 189–211. Deletion of this region from full-length TP53INP2 led to a nuclear–cytoplasmic localization compared to the WT protein, which is predominantly nuclear (Fig. 2B). Moreover, fusing the 189–211 region to EGFP resulted in the complete nuclear localization of EGFP (Fig. 2B). Together with bioinformatics tools and a recent publication that predicted an NLS within the 144–221 region of TP53INP2 (You et al., 2019), our results show that the 189–211 region directs nuclear localization.

Endogenous TP53INP2 is enriched in the nucleoli where it plays a role in ribosome biogenesis. The 191–212 region constitutes a nucleolar localization signal (NoLS) (Xu et al., 2016). This NoLS closely corresponds to the NoLS prediction (amino acids 186–212) we obtained from the nucleolar localization sequence detector server (Scott et al., 2011) (Fig. 2C). In line with this, we observed nucleolar enrichment of EGFP–TP53INP2 both upon transient transfection and in the EGFP–TP53INP2 reconstituted KO cell line (Figs 1A,D, 2B). The consensus sequence based on several proteins with NoLS is designated as R/K-R/K-X-R/K (Weber et al., 2000). Sequence analysis predicted a putative NoLS sequence in the 201–204 region, partially overlapping the NLS. Deletion of the 201–204 region from full-length TP53INP2 prevented localization

of TP53INP2 to the nucleolus (Fig. 2B) and successfully mapped the NoLS of TP53INP2. This NoLS sequence is not conserved in TP53INP1, which is consistent with no nucleolar localization being observed for TP53INP1 (see Fig. 6A,E). Together, our results and previously published data identify overlapping NLS and NoLS signals in the C-terminal region of TP53INP2 (Fig. 2C).

### Deletion of the TP53INP2 NLS leads to accumulation in LC3B-positive cytoplasmic puncta

Exogenously overexpressed TP53INP2 forms puncta in the cytoplasm colocalized with ATG8 proteins upon induction of autophagy by starvation (Nowak et al., 2009; Sancho et al., 2012) (Fig. S3). However, confocal fluorescence microscopy analysis of our TP53INP2 KO cell line reconstituted with EGFP–TP53INP2 expressed at a low, closer to endogenous, level, did not display any cytoplasmic puncta upon starvation (Fig. 1A). Treatment with the lysosomal inhibitor BafA1 did not stabilize EGFP–TP53INP2, whereas the proteasomal inhibitor MG132 caused strong stabilization (Fig. 1B). Thus, overexpression of TP53INP2 likely overloads the regulation mechanisms for nuclear import of TP53INP2 leading to cytoplasmic accumulation and might potentiate TP53INP2 as an autophagic substrate. To clarify this, we generated a HeLa FlpIn cell line with inducible expression of EGFP–TP53INP2 lacking the NLS and NoLS sequences (Δ189–211). Interestingly, deletion of the NLS-NoLS region led to formation of cytoplasmic puncta that colocalized with LC3B under basal conditions. The number of colocalizing cytoplasmic puncta increased upon inhibition of lysosomal degradation and even more upon starvation and inhibition of lysosomal degradation (Fig. 3A,B). Monitoring the expression level of EGFP–TP53INP2(Δ189–211) by western blotting did not reveal a statistically significant accumulation of the NLS-NoLS mutant upon inhibition of lysosomal degradation. Also, the NLS-NoLS mutant was profoundly stabilized by proteasome inhibition (Fig. 3C,D). In contrast to the WT protein, which accumulated in the nucleoli upon proteasomal inhibition, TP53INP2(Δ189–211) accumulated in the cytoplasm and in the nucleoplasm outside the nucleoli. Taken together, our results show that TP53INP2 is predominantly degraded by the proteasome although some of the smaller cytoplasmic pool might be degraded by autophagy.

Deletion of the LIR motif in the TP53INP2(Δ189–211) construct impaired the recruitment of TP53INP2 to the cytoplasmic LC3B dots (EGFP–TP53INP2 ΔNLS+ΔLIR in Fig. 3E), confirming previous studies based on transient overexpression (Nowak et al., 2009; Sancho et al., 2012). These results suggest that TP53INP2 is more stable when localized in the cytoplasm than in the nucleus. This is consistent with published data showing that TP53INP2 is very unstable with a half-life of ∼4 h (Mauvezin et al., 2010). TP53INP2 that has accumulated in the cytoplasm is recruited to LC3B-positive structures via its LIR motif and can undergo degradation via the autophagic-lysosomal pathway as published for TP53INP1 (Seillier et al., 2012). However, normally TP53INP2 is a substrate for proteasomal degradation in the nucleus.

### The starvation-induced cytoplasmic localization of TP53INP2 is independent of ATG5 and NES motifs

TP53INP2 is reported to transport nuclear LC3B to the cytoplasm to promote autophagy under starvation (Huang et al., 2015). In a previous study based on transient transfection and overexpression it was found that translocation of overexpressed TP53INP2 from the nucleus to the cytoplasm upon starvation required a functional LIR motif (Sancho et al., 2012). A nuclear export sequence (NES) was

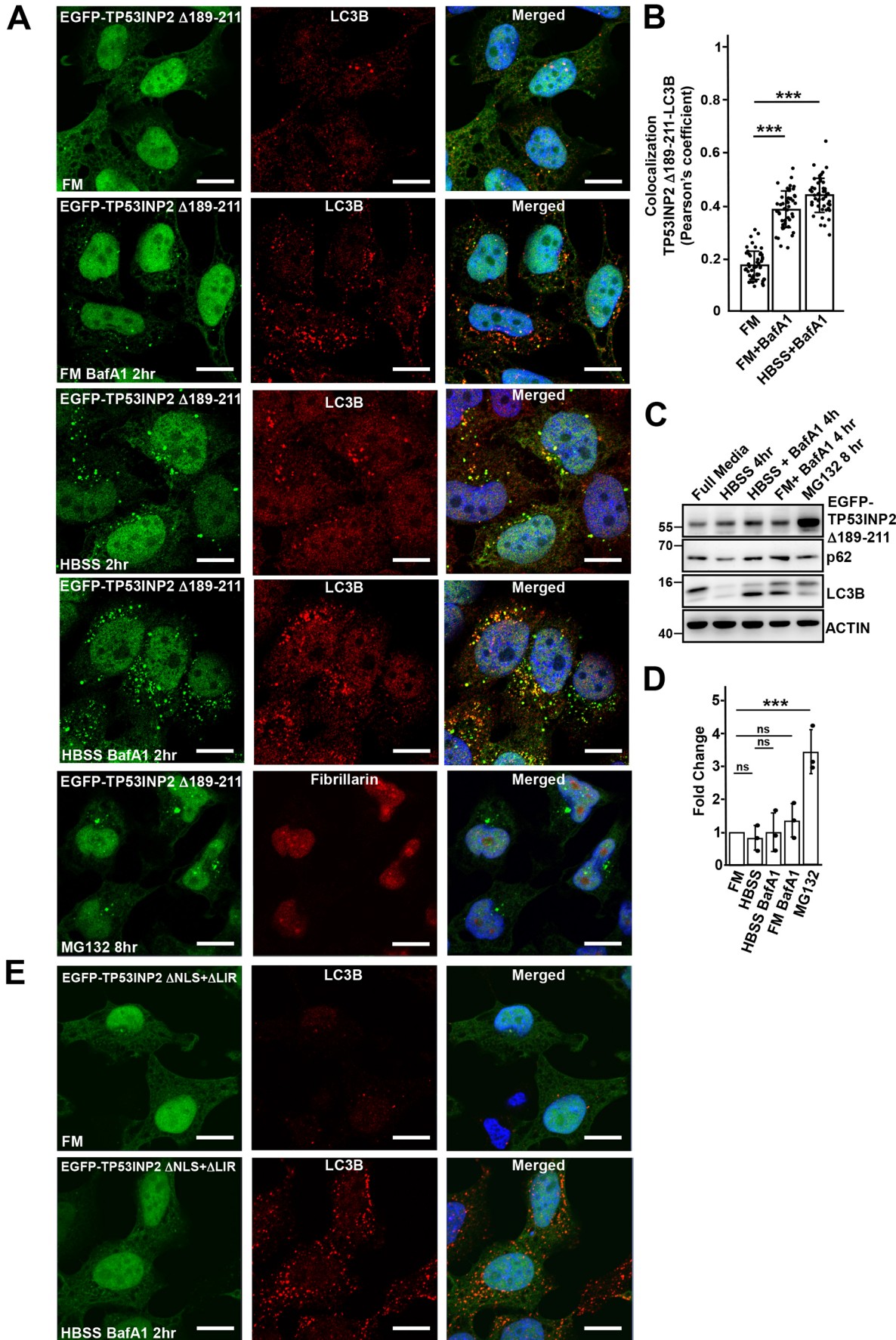

**Fig. 3.** See next page for legend.

**Fig. 3. Deletion of the NLS motif of TP53INP2 leads to cytoplasmic protein accumulation in LC3B-postive puncta.** (A) TP53INP2 Δ189–211 localizes both to the nucleus and cytoplasm and colocalizes with LC3B puncta. After induction of expression of EGFP–TP52INP2 Δ189-211 with 1 µg/ml doxycycline for 24 h, cells were treated as indicated and stained with anti-LC3B or -fibrillarin antibodies, before analysis by confocal microscopy. BafA1 was used at 200 µM for 2 h in HBSS-treated cells where indicated. (B) TP53INP2 Δ189–211 shows increased colocalization with LC3B under starvation and inhibition of lysosomal degradation. The colocalization, expressed as a Pearson's correlation coefficient, are represented as mean±s.d. (*n*=49). ***P<0.001; ns, not significant (one-way ANOVA followed by Tukey's multiple comparison test). (C) After induction of EGFP–TP53INP2 Δ189–211 with doxycycline, cells were treated as indicated and cell extracts analyzed by western blotting using the indicated antibodies. (D) Quantification of the levels of EGFP-TP53INP2 Δ189-211 from three replicates. The bars represent the mean±s.d. of band intensities relative to the actin loading controls. Statistical comparisons were done as for Fig. 1C, F. (E) HeLa cells expressing EGFP TP53INP2 (Δ189–211, Δ24–40) were induced with doxycycline and analyzed by confocal microscopy. Images in A and E representative of three repeats. Scale bars: 10 µm.

identified in the region amino acids 25–47 encompassing the LIR motif at positions 35–38 (Sancho et al., 2012). Together, these studies indicate a mutual role of ATG8 proteins and TP53INP2 in facilitating the export of each other out of the nucleus upon mTOR inactivation. It has been reported that autophagy-deficient cells lacking ATG5 display impaired nucleocytoplasmic shuttling of TP53INP2 (Nowak et al., 2009). To study this more closely, we first applied bioinformatics tools to predict putative NES motifs in TP53INP2. In addition to the published NES1 (25-VSEEDEVDGWLIIDLPDSYAAP-47), another NES2 motif (87-GPARLQSSPLEDLLIEH-104) was predicted. To test their functionality in mediating nuclear export in isolation, both motifs were cloned into the Rev 1.4 plasmid vector (Pankiv et al., 2010). The positive control REV NES motif directed REV–EGFP out of the nucleus (Fig. S4A). However, both the NES1 and NES2 motifs failed to lead to export of REV–EGFP out of the nucleus (Fig. S4B). When HeLa FlpIn EGFP–TP53INP2 cells were starved and treated with leptomycin B (LMB) for 2 h to inhibit NES-dependent nuclear export, EGFP–TP53INP2 was still localized in the cytoplasm and not retained in the nucleus as occurred for p62 (Fig. 4A). p62 continuously shuttles between the nucleus and the cytoplasm (Pankiv et al., 2010). LMB inhibits CRM1 (also known as exportin 1, or XPO1)-mediated export from the nucleus to the cytoplasm. These results show that NES1 and NES2 in TP53INP2 are not functional motifs, and TP53INP2 is not exported out of the nucleus by CRM1. Interestingly, similar to TP53INP2, LMB treatment is not able to restrict LC3B to the nucleus (Drake et al., 2010). Therefore, we analyzed the importance of the LIR motif in TP53INP2 for nucleocytoplasmic shuttling by establishing a HeLa FlpIn cell line expressing EGFP–TP53INP2 with W35A and I38A mutations (W35A/I38A; LIR mutant). Surprisingly, and in contrast to previous studies where transient transfection was used (Sancho et al., 2012), we found that mutation of the LIR motif did not impact TP53INP2 localization (Fig. 4B). Deletion of the complete NES1 encompassing the LIR (TP53INP2 Δ24–40) (Fig. S4C), deletion of the N terminal region comprising 45 amino acid residues (TP53INP2 46–220) (Fig. S4D) and mutation of residues E97 and D98, previously reported to be important for TP53INP2 shuttling (Sancho et al., 2012), did not affect localization either (Fig. S4E). Hence, nuclear export of TP53INP2 seems not to be mediated by the previously published NES motif or by interaction with the ATG8 proteins.

We next asked whether ATG5 and autophagy activity impacted on TP53INP2 localization. To this end, we generated WT mouse embryonic fibroblast (MEF) and ATG5 KO MEF cell lines constitutively expressing EGFP–TP53INP2. In contrast to a previous report based on transient transfection and overexpression (Nowak et al., 2009), the ATG5 KO MEF EGFP–TP53INP2 cell line and WT MEF cells both showed the same nuclear localization of EGFP–TP53INP2 in FM and cytoplasmic localization in HBSS (Fig. 4C,D). Hence, autophagy does not impact directly on the subcellular localization of TP53INP2. Together, our results suggest that at an expression level much closer to the endogenous level of expression of TP53INP2 than obtained by transient overexpression, the protein is not shuttling between the nucleus and the cytoplasm.

## Starvation induces cytoplasmic retention of newly synthesized TP53INP2 that is not regulated by post translational modifications

Our results so far raised the question of whether the cytoplasmic accumulation of TP53INP2 observed upon starvation is regulated by inhibition of nuclear import and not by nuclear export. To investigate this, we performed fluorescence recovery after photobleaching (FRAP) on the TP53INP2 KO cell line reconstituted with EGFP–TP53INP2. Nuclear EGFP–TP53INP2 was completely photobleached and the cells were starved in HBSS for 1 h. During this period, the localization of EGFP–TP53INP2 was followed by live-cell imaging. Despite photobleaching of the entire nuclear pool of EGFP–TP53INP2, we still observed cytoplasmic accumulation similar to what was seen in the unbleached cells, upon starvation (Fig. 5A). Thus, the cytoplasmic accumulation takes place without contribution from the nuclear pool of EGFP–TP53INP2. Upon inhibiting protein synthesis with cycloheximide we found that the cytoplasmic accumulation of TP53INP2 upon starvation is dependent on new protein synthesis (Fig. 5B,C). We therefore hypothesized that newly synthesized EGFP–TP53INP2 is restricted to the cytoplasm upon starvation, whereas the nuclear EGFP–TP53INP2 pool is degraded by nuclear proteasomes. To test this further, we checked whether the interaction between TP53INP2 and the importin family of proteins is modulated upon starvation. For nuclear import, proteins interact with the nuclear pore complex whereby the importin family of proteins plays an essential role (Pankiv et al., 2010). First, we assayed the interaction of TP53INP2 with importin family proteins by GST pulldown assay (Fig. 5D). As TP53INP2 bound best to importin α1, GST–importin α1 was applied in a GST-pulldown assay with cell lysates from the HeLa FlpIn EGFP–TP53INP2 cell line exposed to FM or HBSS for 2 h. Interestingly, EGFP–TP53INP2 from cells exposed to FM interacted strongly with GST–importin α1 whereas EGFP–TP53INP2 from cells exposed to HBSS did not (Fig. 5E). This indicates that HBSS treatment affects the binding of TP53INP2 to Importin and thereby restricts TP53INP2 from entering the nucleus, keeping it in the cytoplasm.

Acetylation of proteins can regulate their subcellular localization (Narita et al., 2019). Importantly, nuclear localization of LC3B is suggested to be regulated via acetylation of the lysine residues K49 and K51 (Huang et al., 2015). This prompted us to investigate whether starvation could change the acetylation pattern of TP53INP2 and thereby inhibit importin binding and nuclear translocation. To this end, we mutated each of the three individual lysine residues within TP53INP2, of which two are localized inside and next to the NLS motif, to arginine residues. We tested both single mutations of the three lysine residues and mutations of all

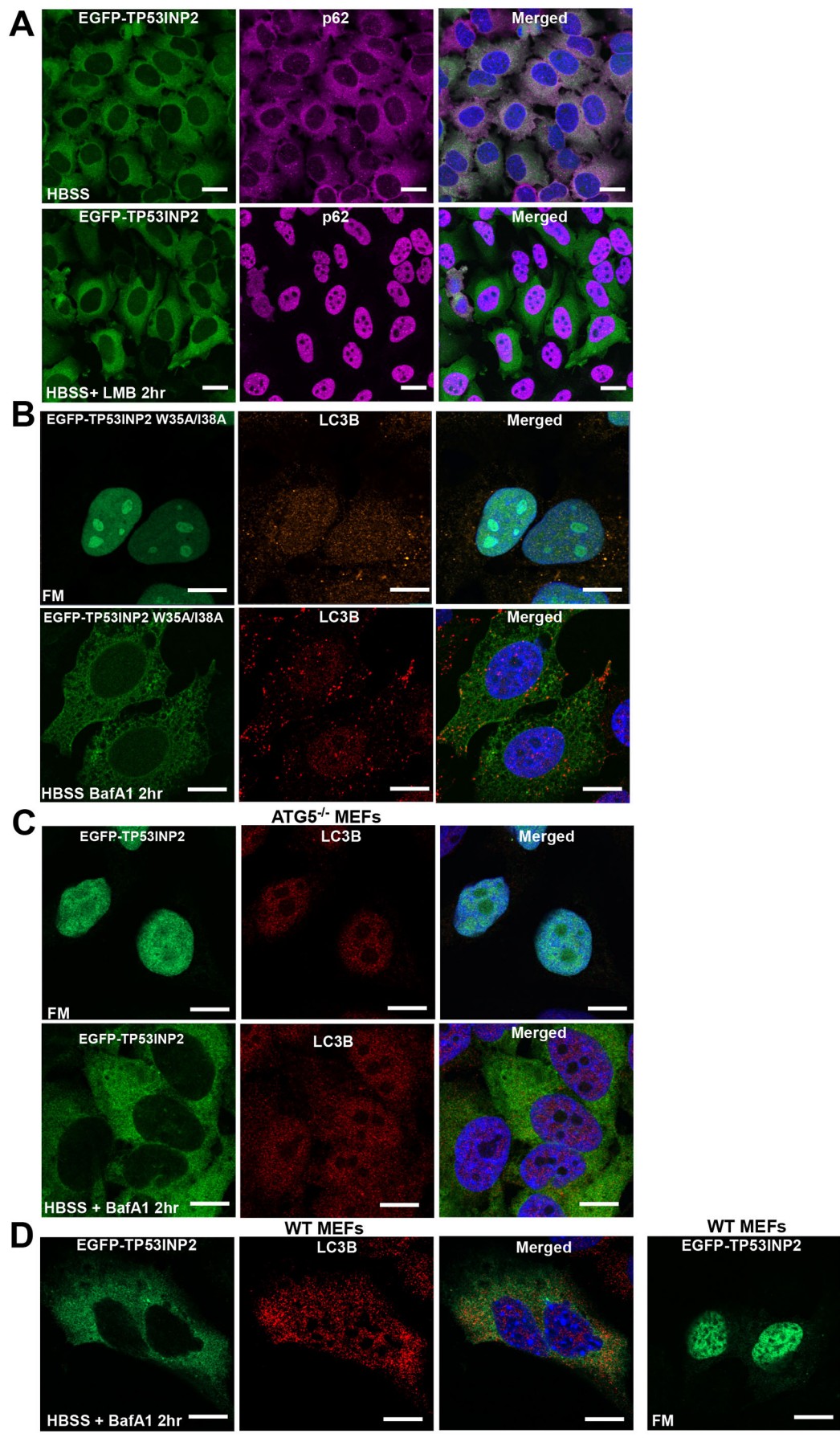

**Fig. 4.** See next page for legend.

**Fig. 4. The starvation-induced cytoplasmic localization of TP53INP2 is independent of NES and LIR motifs.** (A) Treatment of cells with the nuclear export inhibitor leptomycin B (LMB, 50 nM for 2 h) does not inhibit the nuclear to cytoplasmic redistribution of TP53INP2 in response to starvation (HBSS). HeLa FlpIn cells expressing EGFP–TP53INP2 were treated as indicated and stained for endogenous p62 as a positive control for LMB treatment, before analysis by confocal imaging. (B) EGFP–TP53INP2 with a mutated LIR motif (W35A/I38A) shows similar cytoplasmic localization to that in WT in response to starvation. HeLa FlpIn cells expressing EGFP–TP53INP2 W35A/I38A were treated as indicated and stained with anti-LC3B antibodies, before analysis by confocal imaging. (C,D) Stable WT MEF and ATG5$^{-/-}$ MEF cell lines expressing EGFP–TP53INP2 made by retroviral transfer were treated with 1 μg/ml doxycyline for 24 h to induce expression of EGFP–TP53INP2. Cells were then treated as indicated and stained with p62 and LC3B antibodies, before analysis by confocal fluorescence microscopy. BafA1 was used at 200 μM for 2 h. Images are representative of three repeats. Scale bars: 10 μm.

three lysine residues together (Fig. 5F), and all did not affect the starvation-induced cytoplasmic retention. This suggests that during mTOR inactivation, acetylation does not negatively regulate nuclear import of TP53INP2. To rule out the possibility of phosphorylation regulating subcellular localization, we made several cell lines expressing EGFP–TP53INP2 with deletions (Fig. S5A–F) or point mutations of serine-threonine phosphorylatable residues (Fig. S5G–K), but we could not see any effect of any of these mutations on subcellular localization.

## Cytoplasmic retention of TP53INP2 and TP53INP1 is regulated by an evolutionary conserved C-terminal 9-amino-acid CRM

*TP53INP1* and *TP53INP2* arose by gene duplication of an ancestral gene that encoded a protein most similar to TP53INP2 in the lineage before vertebrates evolved (Sancho et al., 2012). The human proteins show 30% sequence identity with a strong conservation of the nine-most C-terminal amino acids with seven out of nine being identical between TP53INP1 and TP53INP2 (Fig. 6A). We subjected the C-terminal amino acid sequences of TP53INP1 and 2 to analyses performed by the ConSurf server. These analyses projects evolutionary conservation scores onto the 3D structures of proteins to map functional sites (Ashkenazy et al., 2016; Yariv et al., 2023). This shows that six of the nine-most C terminal residues are highly conserved and predicted to be exposed and functionally important (Fig. 6B,C). Deletion of the conserved C-terminal 9 amino acids of both TP53INP1 and TP53INP2 showed the same nuclear translocation as full-length protein in FM (Fig. 6D,E). As shown for TP53INP2 (Figs 4B, 5D–G), mutation of individual lysine residues and LIR motif within TP53INP1 gave rise to a similar cytoplasmic retention under starvation as wild type (Fig. S6A,B). However, during starvation (HBSS), both TP53INP1 and TP53INP2 lacking this C-terminal region showed reduced cytoplasmic retention compared to that for full-length proteins (Fig. 6D,E). Upon starvation, the evolutionary conserved C-terminal motif of TP53INP1 and TP53INP2 might interact with protein(s) that inhibit the interaction with importin to block the nuclear import of TP53INP1 and 2. We call this motif a cytoplasmic retention motif (CRM). To test the CRM further, we made two cell lines with point mutations in the CRM by stably reconstituting HeLa FlpIn TP53INP2 KO cells with either the EGFP–TP53INP2 Q212A/P213A double mutant or the R216A/Q217A double mutant (Figs 6C, 7A). Q212, P213 and R216 are completely conserved whereas Q217 is less conserved in evolution (Fig. 6B). Consistent with this, the Q212A/P213A mutant had the same dramatic effect on loss of cytoplasmic retention as the deletion of the

entire CRM whereas the R216A/Q217A mutant had a much less severe effect (Fig. 7A).

## Increased translation of TP53INP2 upon starvation is mediated by the m$^6$A mRNA demethylase FTO

Having established that incubation in starvation medium (HBSS) leads to cytoplasmic retention of both TP53INP1 and 2 via the conserved C-terminal 9-amino-acid CRM we asked whether also other mechanisms could contribute to the increased cytoplasmic level. The expression of the m$^6$A mRNA demethylase fat mass and obesity-associated protein (FTO) is increased by metabolic stress conditions, such as low serum (0.2% FBS) or starvation medium (HBSS) (Yang et al., 2019). Recently, a study showed that the expression of TP53INP2 was upregulated by FTO (Huang et al., 2023). We therefore set out to test whether FTO could contribute to increased cytoplasmic levels of TP53INP2 upon starvation. Using the FTO demethylase inhibitor FB23-2, the GFP fluorescence intensity of EGFP–TP53INP2 in the cytoplasm was strongly reduced upon starvation as was the protein levels analyzed by western blotting (Fig. 7B–D). siRNA-mediated knockdown of FTO also inhibited the starvation-induced increased levels of TP53INP2 (Fig. 7E,F). The protein level of EGFP–TP53INP2 increased in a time series of starvation (Fig. 7G). However, quantitative (q)PCR experiments in this starvation time series showed no significant changes in the mRNA levels of *EGFP–TP53INP2* (Fig. 7H). Hence, the results indicate that FTO does not mediate direct stabilization of *TP53INP2* mRNA but rather involves increased translation of *TP53INP2* mRNA. The inhibition of FTO is known to inhibit translation efficiency of E2F1 and Myc in HeLa cells (Zou et al., 2019).

Our data allows us to propose the following model (Fig. 7I). In nutrient-replete conditions without cell stressors present, *TP53INP2* mRNA is likely partially demethylated by FTO. The protein is imported into the nucleus where it is degraded by the proteasome after having acted as a transcriptional cofactor. Upon acute nutrient starvation FTO is more active and demethylates *TP53INP2* mRNA resulting in increased TP53INP2 protein. However, the protein is not imported into the nucleus but is retained in the cytoplasm by a protein or proteins binding to the C-terminal CRM.

## DISCUSSION

The major conclusion of this study is that the localization switch of both TP53INP1 and TP53INP2 upon nutrient starvation and other stresses from the nucleus to the cytoplasm is regulated by a conserved 9-amino-acid CRM at their C termini. By establishing TP53INP2 KO cell lines with low expression levels of EGFP–TP53INP2, we found that EGFP–TP53INP2 does not shuttle from the nucleus to the cytoplasm. Instead, mTOR inhibition by nutrient starvation led to an FTO-mediated increase in TP53INP2 protein, which is retained in the cytoplasm by the CRM. Within the time frame studied here the cytoplasmic TP53INP2 protein is not degraded by autophagy although nutrient starvation induces autophagy. Under nutrient starvation and several other stress conditions the cytoplasmic pool of TP53INP2 is not free to be imported actively by importins, or to passively diffuse, into the nucleus to be degraded by proteasomes there.

Data presented in a previous study suggested that TP53INP2 shuttles between the cytoplasm and nucleus. During this shuttling, it passes through the nucleolus (Mauvezin et al., 2012). TP53INP2 has also been reported to act as a mediator of nucleocytoplasmic shuttling of LC3B, binding to deacetylated LC3B in the nucleus and facilitating its export to the cytoplasm upon nutrient deprivation (Huang et al., 2015). Our data does not oppose that nuclear export of

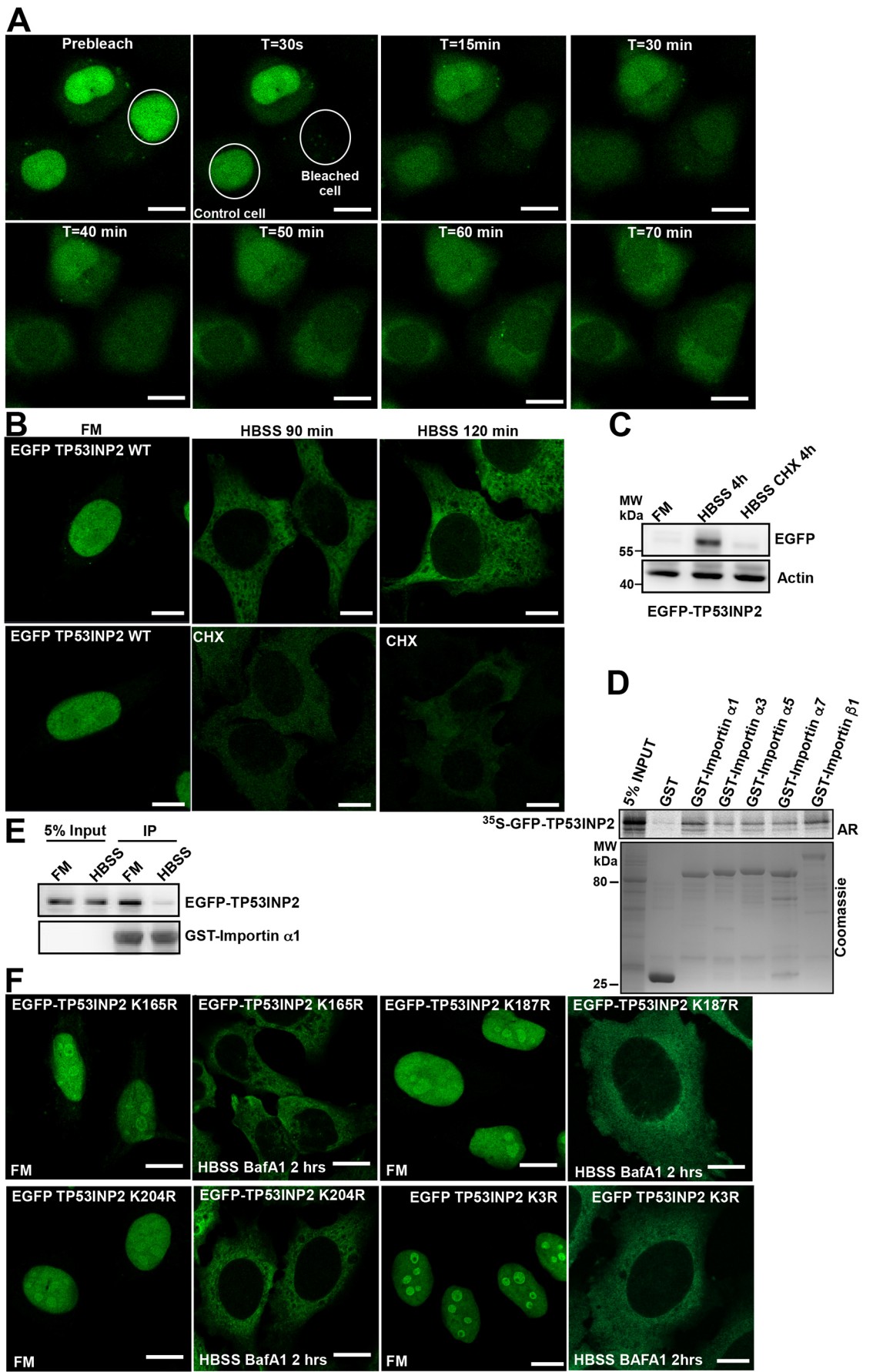

**Fig. 5.** See next page for legend.

**Fig. 5. Starvation induces cytoplasmic retention of newly synthesized TP53INP2 independently of acetylation.** (A) Cytoplasmic accumulation of TP53INP2 upon starvation does not depend on the nuclear pool of TP53INP2. Confocal fluorescence microscopy live-cell imaging of HeLa FlpIn cells expressing EGFP–TP53INP2 under starvation. The nucleus of one of the representative cells was photobleached while another cell was left unbleached. After complete photobleaching of nuclear EGFP–TP53INP2, cells were treated with HBSS and live-cell imaging performed. (B,C) Cycloheximide blocks new synthesis of TP53INP2 during starvation. Confocal fluorescence imaging of HeLa FlpIn cells expressing EGFP–TP53INP2 under starvation treated or not with 50 µg/ml of cycloheximide (CHX) for 90 or 120 min (B). Western blot of EGFP–TP53INP2 in FM, 4 h in HBSS and 4 h in HBSS with 50 µg/ml CHX (C). (D) TP53INP2 interacts with importin α1. GFP-tagged TP53INP2 was *in vitro* translated in the presence of [$^{35}$S]methionine and analyzed in GST pulldown assays for binding to the indicated recombinant importin family proteins fused to GST. Bound proteins were detected by autoradiography (AR), and immobilized GST fusion proteins by Coomassie Brilliant Blue staining. (E) The binding of TP53INP2 to importin α1 is reduced upon starvation. HeLa FlpIn cells expressing EGFP–TP53INP2 were either grown under full medium (FM) or starved with HBSS for 2 h, and cell lysates tested in GST pulldown assays for binding of EGFP–TP53INP2 to GST–importin α1. Bound EGFP–TP53INP2 was detected with anti-EGFP antibodies and immobilized GST–importin α1 by Coomassie Brilliant Blue staining. (F) HeLa cells expressing EGFP–TP53INP2 K165R, K187R, K204R and K3R (K165R, K187R and K204R) were either grown in full medium or HBSS for 2 h. Mutation of all three lysine residue had no effect on cytoplasmic retention during amino acid starvation. Images in A, B and F representative of three repeats. Blots in C–E representative of two repeats. Scale bars: 10 µm.

LC3B occurs upon mTOR inhibition. However, TP53INP2 is not exported together with LC3B. This conclusion is based on (1) lack of a functional NES in TP53INP2, (2) the inability of the CRM1 inhibitor Leptomycin B to restrict TP53INP2 to the nucleus, and (3) by FRAP data showing that TP53INP2 does not move out of the nucleus. Our data suggests a model where mTOR inhibition leads to stabilization of the cytoplasmic TP53INP2 while the nuclear pool of TP53INP2 is degraded by the proteasome (Fig. 7I). The interaction of the CRM with a cytoplasmic component(s) upon starvation precludes TP53INP1 and 2 from interacting with LC3B in the cytoplasm. When the CRM is deleted, there is a lot of colocalization of TP53INP2 with LC3B (and nuclear import), like what is seen when TP53INP2 is overexpressed by transient transfection.

In nutrient-replete conditions TP53INP2 is localized in the nucleus and enriched in the nucleolus. It acts both as a transcriptional coactivator of the thyroid hormone receptor and as a facilitator of RNA polymerase I transcription (Baumgartner et al., 2007; Xu et al., 2016). These findings were supported by our analysis of the TP53INP2 KO cells, which displayed a lower proliferation rate and less rDNA transcription than the WT cells. We mapped the NLS of TP53INP2 to amino acids 189–211. Within this region, we mapped the NoLS to amino acids 201–204. Interestingly, mutation of all three lysine residues to arginine residues did not inhibit the starvation-induced cytoplasmic retention ruling out the possibility of acetylation mediated regulation. The possibility of phosphorylation-mediated regulation was also ruled out by deletion or point mutation of serine and threonine residues in TP53INP2. The highly conserved C terminal 9-amino-acid CRM in TP53INP1 and 2 regulates the nuclear import upon nutrient-depleted conditions. When the nuclear import of TP53INP2 is inhibited under nutrient deprivation its stimulating effects on cell proliferation and ribosome biogenesis are consequently blocked. The nuclear fraction of TP53INP2 is rapidly degraded by proteasomes to ensure that the anabolic roles of TP53INP2 are switched off during starvation (Fig. 7I).

Other nuclear proteins are also reported to be retained or translocated in the cytoplasm upon nutrient starvation. Foxk1 and 2 are repressors of important autophagy genes under nutrient-rich conditions. Upon nutrient starvation the Foxk proteins are re-located from the nucleus to the cytoplasm. The inhibition of mTORC1 prevents nuclear import of these proteins under starvation conditions (Bowman et al., 2014). Amino acid starvation promotes the translocation of the 26S proteasome from the nucleus to the cytosol. Removal of the aromatic amino acids sensed by sestrin 3 leads to mTOR inhibition and accumulation of proteasomes in the cytosol (Livneh et al., 2023). Proteasomes, now in the cytosol, lead to proteolysis to replenish amino acids for cell survival. Starvation in HBSS (and oxidative stress) causes high mobility group box 1 (HMGB1), a chromatin-associated nuclear protein and extracellular damage-associated molecular pattern molecule, to exit the nucleus and accumulate in the cytoplasm, where it binds beclin 1 to stimulate autophagy (Tang et al., 2010). The Hippo pathway serine/threonine protein kinase STK38 accumulates in the cytoplasm upon amino acid starvation and by phosphorylating CRM1 STK38 mediates nuclear export of itself, the autophagy protein beclin 1 and the Hippo pathway coactivator YAP1 (Martin et al., 2019). The carbohydrate-response element-binding protein (ChREBP; also known as MLXIPL) is a glucose-responsive transcription factor that plays an essential role in converting excess carbohydrates to fat storage in the liver. Upon glucose starvation ChREBP is phosphorylated and retained in the cytoplasm by interaction with 14-3-3 proteins with AMP activating this interaction. This leads to cytoplasmic retention because the NLS cannot be accessed by importins (Sato et al., 2016). For TP53INP2, proteomics experiments done in our lab have so far not enabled us to identify which protein(s) are responsible for the cytoplasmic retention of TP53INP2.

We found the protein level of TP53INP2 to be strongly increased upon starvation. Treatment with cycloheximide showed that this increase is dependent on protein synthesis, and not only due to protein stabilization. Thus, TP53INP2 escapes the global downregulation of translation that occurs upon mTOR inhibition. Interestingly, TP53 is shown to be stabilized by starvation in hepatic tissue (Prokesch et al., 2017). The expression of the m6A mRNA demethylase FTO is increased upon starvation (Yang et al., 2019), and the level of TP53INP2 is upregulated by FTO (Huang et al., 2023). Treatment with FTO inhibitor markedly reduced cytoplasmic EGFP–TP53INP2 fluorescence and total protein levels upon starvation. siRNA-mediated knockdown of FTO suppressed starvation-induced TP53INP2 upregulation. TP53INP2 protein levels increased over time during starvation, but qPCR revealed no significant change in mRNA levels, suggesting that FTO enhances TP53INP2 expression by increasing translation efficiency rather than mRNA stabilization. This correlates with observations of reduced translation efficiency of transcription factors E2F1 and Myc in HeLa cells when FTO is inhibited (Zou et al., 2019). Another study reported increased translation of TP53INP1 during TNF-α treatment, involving m6A methylation of its mRNA by the methyltransferase METTL3, and subsequently a transition of its mRNA from a non-polysomal to a polysomal fraction (Akcaoz-Alasar et al., 2024). The role of starvation-induced TP53INP2 expression will be an important question to address in future studies.

Cytoplasmic TP53INP2 is reported to facilitate autophagosome formation (Nowak et al., 2009). Whether TP53INP2, via binding to the vacuole membrane protein 1 (VMP1), functions as a scaffold for LC3B recruitment to autophagosomes, or whether LC3B recruits TP53INP2 to the phagophore structures, has not been clarified

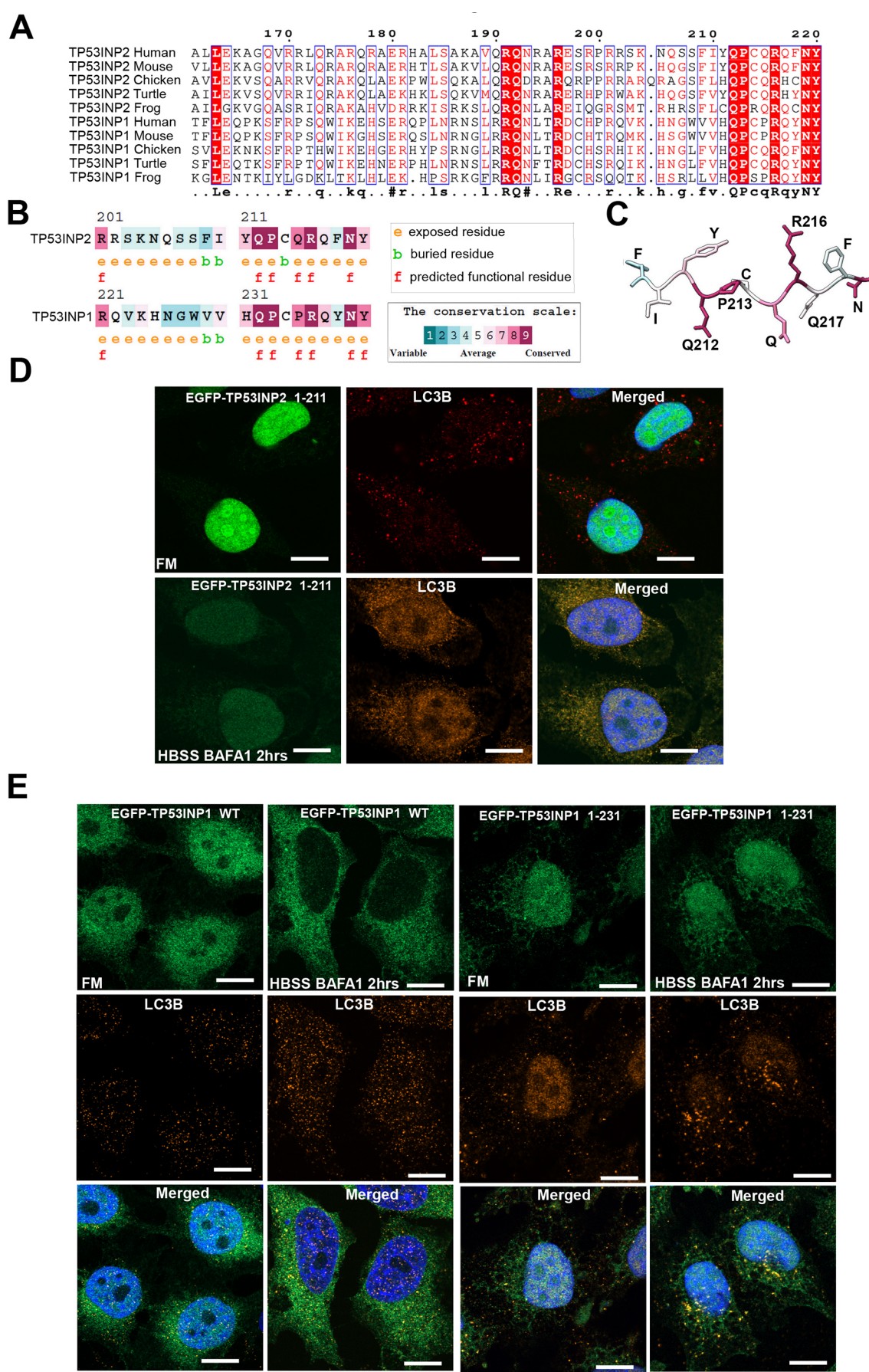

**Fig. 6.** See next page for legend.

**Fig. 6. Cytoplasmic retention of TP53INP1 and TP53INP2 are regulated by the C-terminal region.** (A) Multiple sequence alignment of TP53INP1 and TP53INP2 across different species. (B,C) The analysis of C-terminal amino acid sequence of TP53INP1 and 2 by ConSurf server showing that the region is highly conserved, exposed and functional. (D,E) HeLa FlpIn cells expressing EGFP–TP53INP2 1–211 and TP53INP1 1–231 showed reduced cytoplasmic retention compared to full-length protein during starvation. BafA1 was used at 200 μM for 2 h. Images in D and E representative of three repeats. Scale bars: 10 μm.

(Huang et al., 2015; Mauvezin et al., 2012; Nowak et al., 2009; Ropolo et al., 2007). However, a recent report shows that LC3B recruits TP53INP2 to autophagic membranes (You et al., 2019), which is in line with our results. Our data suggests that cytoplasmic retention of TP53INP2 is regulated by its CRM. This regulation is overridden when TP53INP2 is overexpressed by transient transfection. Then it is recruited to LC3B-positive structures in an LIR-dependent manner. However, when the expression of TP53INP2 is kept at a level closer to the endogenous level, TP53INP2 is stabilized in the cytoplasm upon starvation and not detected in LC3B-positive structures. Normally, TP53INP2 is degraded by the proteasome (Mauvezin et al., 2010), which we show occurs in the nucleus, most likely in the nucleolus, through a ubiquitin-independent pathway. Similar to our result with TP53INP2, PICT1 (NOP53), a nucleolar ribosomal protein is also degraded inside the nucleolus in a ubiquitin-independent manner (Maehama et al., 2014). This degradation pathway is not well understood. PICT1, a regulator of p53, has been found to be degraded by a nucleolar ubiquitin-independent proteasomal pathway (Maehama et al., 2014). Whether TP53INP2 is degraded by the same pathway is an interesting question to address in future studies.

Our measurements of autophagic flux in the TP53INP2 KO out HeLa cells revealed no difference from WT cells. LC3B lipidation upon starvation was normal, and the autophagy receptors did not accumulate. Thus, loss of TP53INP2 does not affect autophagy flux detectably in this cancer cell line. Another recent study reported that TP53INP2 interaction with ATG7 enhanced LC3–ATG7 interaction but did not affect the formation of autophagosomes on knockdown of TP53INP2 (You et al., 2019). Consistent with this, TP53INP2-deficient 3T3-L1 cells did not display any changes in autophagy flux during adipogenesis relative to control cells (Romero et al., 2018). Together, these findings raise the question of whether TP53INP2 might have other roles in the cytoplasm rather than being a regulator of autophagy activity. Interestingly, a recent paper shows that TP53INP2 sensitizes cells to apoptosis induced by death receptor ligands (Ivanova et al., 2019). TP53INP2 was found to bind to caspase-8 and the ubiquitin ligase TRAF6 and thereby regulate TRAF6-mediated activation of caspase-8. Hence, an important role of TP53INP2 in the cytoplasm might be to regulate this death receptor pathway. To conclude, TP53INP2 serves important roles in the cytoplasm, nucleus and nucleolus. We now show that its subcellular localization is tightly regulated by its conserved C-terminal CRM.

## MATERIALS AND METHODS
### Plasmids
The Gateway entry clones used in this study are listed in Table S1. A QuikChange site-directed mutagenesis kit (Stratagene) was used to create desired point mutation which was verified by DNA sequencing (BigDye sequencing kits, Applied Biosystems). For generation of Gateway destination plasmids, the Gateway LR and BP recombination kit from Invitrogen was used.

### Cell culture
HeLa Flp-In T-REx cells (Invitrogen, R714-07) were cultured in Eagle's minimum essential medium with 10% serum (Biochrom, S0615) and 1% streptomycin-penicillin (Sigma-Aldrich, P4333) (denoted as full medium, FM, in the text) or Hanks' balanced salt solution (Sigma-Aldrich, H9269) (denoted as HBSS in the text). All HeLa FlpIn cell lines with EGFP-TP53INP2 integrated at FRT site were maintained in the same medium with additional selection antibiotics 100 μg/ml hygromycin (Calbiochem, 400051) and 7.5 μg/ml blasticidin S (Invitrogen, R210-01).

### Generation of stable cell lines
HeLa FlpIn T-REx cells were used to make stable TP53INP2 cell lines. The N-terminal EGFP-tagged TP53INP2 cDNAs were cloned into pcDNA5 FRT/TO plasmid. The generation of stable cell lines was performed according to the manufacturer's instructions (Invitrogen, V6520-20). Briefly, at 48 h after the transfection of different mutants of TP53INP2 cloned into pcDNA5 FRT/TO plasmids, colonies of cells with the gene of interest integrated into FRT site were selected with 200 μg/ml of hygromycin (Calbiochem, 400051). Gene expression was induced with 1 μg/ml of doxycycline for 24 h.

ATG5 KO and WT MEF cells were kindly provided by Noboru Mizushima (Hosokawa et al., 2006). Stable MEF cell lines expressing EGFP–TP53INP2 cell lines were made by retroviral transfer. First, Platinium Retroviral Packaging Cell Line-E (Cell Biolabs, RV-101) was transfected with a pMXs retroviral plasmid containing EGFP–TP53INP2 cDNA. After 24 h of transfection, viral supernatant was harvested, filtered through a 0.45 μM filter and mixed with 8 μg/ml of Polybrene (Sigma-Aldrich, H9268), before being added to cells. The viral transduction procedure was repeated for the next 48- and 72-h post transfection. After the last viral transduction, MEF cells were selected with 5 μg/ml of blasticidin S (Thermo Fisher Scientific R2210-01).

### CRISPR/Cas9
To construct the specific TP53INP2 guide RNAs the CRISPR/Cas9 plasmid, sense- and antisense oligonucleotides encoding the selection guide sequence were annealed and then inserted into plasmid pSpCas9(BB)-2A-Puro (PX459) vector (Addgene #62988). The target guide sequences were designed using CHOPCHOP web tool (https://chopchop.cbu.uib.no; Labun et al., 2019). For generation of CRISPR/Cas9 KO cells, ~30,000 HeLa Flp-In T-Rex cells were seeded into 24-well plates and then 500 ng of plasmid PX459 were transfected per well using Metafectene Pro (Biontex, T040). The clonal selection was achieved by 500 ng/ml puromycin treatment, 24 h after transfection for 48–72 h. Later, single cells were sorted into 96-well plates via FACS sorting. The clones were grown for 7–10 days and each clone was screened for KO by DNA sequencing of PCR products amplified from the targeted region in the genome.

### Immunofluorescence
For fixed cell imaging, cells were seeded on glass coverslip (VWR, #631–0150) in 24-well plates and for live-cell imaging, cells were grown on Lab-Tek chambered coverglass (Thermo Fisher Scientific, cat. no. 155411). After the indicated treatment, cells were fixed in 4% paraformaldehyde at room temperature for 10 min. Then, cells were washed five times with PBS followed by permeabilization with 0.1% Triton X-100 in PBS for 10 min at room temperature. Subsequently, cells were washed five times with PBS followed by blocking with 5% BSA for 60 min at room temperature. Cells were then incubated with primary antibody for 1 h at room temperature. After washing five times with PBS, cells were incubated with Alexa Fluor-conjugated secondary antibody for 30 min at room temperature followed by washing with PBS five times. Finally, cells on the coverslips were mounted on a glass slide using VECTASHIELD antifade mounting medium (Vector Laboratories, H-1700). Imaging was done with the LSM780 or LSM800 system (Carl Zeiss Microscopy).

### Fluorescence recovery after photobleaching
HeLa Flp-In cells expressing EGFP-TP53INP2 WT or various mutants were grown on Lab-Tek chambered coverglass (Thermo Fisher Scientific, cat.no. 155411) and imaged at 37° C and 5% $CO_2$ on an LSM780 confocal microscope (Carl Zeiss Microscopy) equipped with a 40×1.2 NA water immersion lens. FRAP analysis was performed by drawing regions of

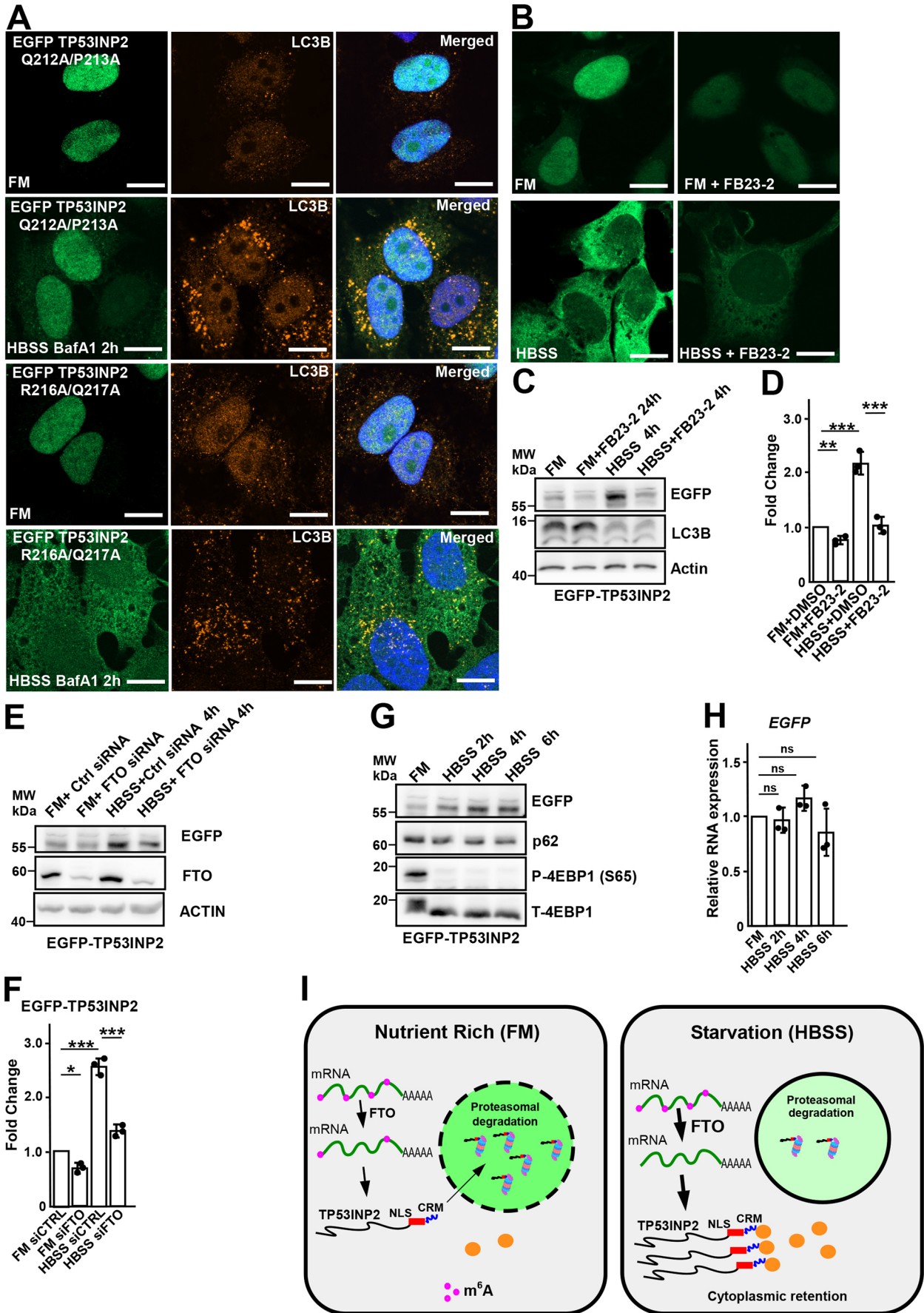

**Fig. 7.** See next page for legend.

**Fig. 7. The starvation induced TP53INP2 mRNA stabilization is mediated by demethylase FTO.** (A) HeLa FlpIn cells expressing EGFP–TP53INP2 Q212A/P213A and TP53INP2 R216A/Q217A were induced with 1 µg/ml doxycycline for 24 h, and grown in full medium (FM) or HBSS for 2 h. During starvation, both cell lines with mutations in the C-terminus showed reduced cytoplasmic retention compared to wild type. BafA1 was used at 200 µM for 2 h. (B,C) HeLa FlipIn cells expressing EGFP–TP53INP2 were induced with doxycycline, and either grown in full medium with or without 20 µM FTO inhibitor FB23-2 for 24 h or starved in HBSS medium with or without 20 µM FTO inhibitor FB23-2 for 4 h. Cells were either analyzed by confocal microscopy (B), or lysed and analyzed by western blotting using the indicated antibodies (C). In both full medium and starvation conditions, the use of FTO inhibitors FB23-2 resulted in reduced fluorescence and protein band intensity of EGFP–TP53INP2. (D) Quantification of immunoblot analysis from three replicates. The bars represent the mean±s.d. of band intensities relative to the actin loading controls. (E) Knockdown of FTO results in lower TP53INP2 levels in both full medium and starvation. HeLa FlpIn cells expressing EGFP–TP53INP2 were transfected with Ctrl siRNA or FTO siRNA followed by 1 µg/ml doxycyline induction 24 h post transfection. Then 48 h post transfection, cells were either grown in full medium or HBSS medium for 4 h, lysed and analyzed by western blotting using the indicated antibodies. (F) Quantification of immunoblot analysis from three replicates. The bars represent the mean±s.d. of band intensities relative to the actin loading controls. (G,H) HeLa FlpIn cells stably expressing EGFP–TP53INP2 were grown in full medium or HBSS for 2 h, 4 h and 6 h. The starvation at different time points showed increased EGFP–TP53INP2 protein band intensities but EGFP–TP53INP2 mRNA levels were unchanged, as measured by RT-PCR with EGFP primer. Results are mean±s.d. ($n$=3). ***$P$<0.001, **$P$<0.005; *$P$<0.01; ns, not significant (one-way ANOVA followed by Tukey's multiple comparison). Images in A and B representative of three repeats. Scale bars: 10 µm. (I) Illustrative model showing C-terminal cytoplasmic retention motif and nuclear localization signal regulating nuclear import of TP53INP2, and FTO-mediated regulation of TP53INP2 expression during culture in nutrient-rich medium and upon starvation.

interest (ROIs) around the nucleus and photobleaching the GFP signal inside the ROI to 100% of its initial value using 5 iterations of unattenuated 488 nm laser light. Fluorescence recovery was then monitored. An ROI placed inside a neighboring cell was monitored to control for photobleaching during image acquisition.

### Antibodies and reagents
Antibodies used in this study are detailed in Tables S2 and S3. DAPI (Thermo Scientific, Cat.no. 62248) was used for nuclear staining.

### Chemicals
The following chemical reagents were used: Bafilomycin A1 (Sigma, B1793), MG132 (Sigma, C2211), Torin 1 (Santa Cruz Biotechnology, sc-396760), CCCP (Sigma, C2759), SFN (Sigma, S4441), thapsigargin (Sigma, T9033), puromycin (Sigma, P8833), sodium arsenite (Sigma, S7400), PP242 (MedChemExpress, HY-10474) Leptomycin B (Sigma, L2913), [$^{35}$S]-methionine (PerkinElmer, NEG709A500UC) and FB23-2 (MedChemExpress, HY-127103). Concentration and duration of use are provided in figure legends when appropriate.

### Real-time PCR
Total RNA was isolated using GenElute Mammalian Total RNA Miniprep Kit (Sigma, RTN70) and reverse-transcribed using Transcriptor Universal cDNA master mix (Roche, cat. no. 05893151001). The real time qPCR was performed using FastStart Universal SYBR Green Master Mix (Roche, Cat.no. 04913850001) on a LightCycler® 96 Real-Time PCR system (Roche). The primers used were as follows: 47S rRNA, forward primer 5′-TGTCAGGCGTTCTCGTCTC-3′, reverse primer 5′-GAGAGCACGA-CGTCACCAC-3′; actin, forward primer 5′-TGACGGTCAGGTCATCA-CTATCGGCAATGA-3′, reverse primer 5′-TTGATCTTCATGGTGAT-AGGAGCGAGGGCA-3′; TP53INP2, forward primer 5′-CCTCCCCTT-CTCCTCCAGTAAA-3′, reverse primer 5′-AGCCCAAAATTCAGTCT-CACCA-3′; and EGFP, forward primer 5′-AGTCCGCCCTGAGCAA-AGA, reverse primer 5′-TCCAGCAGGACCATGTGATC-3′.

### Cell proliferation assay
HeLa FlpIn TP53INP2 KO cells and HeLa FlpIn T-REx control cells were seeded in two parallels of four different densities (2000, 4000, 5000 and 6000 cells) in 100 µl DMEM (Sigma, D6046) on E-Plate L16 PET readers (ACEA Biosciences Inc, #2801185). xCELLigence® Real-Time Cell Analysis (RTCA; ACEA Biosciences) was used to measure the cell proliferation over a time of 96 h with recording at 1 h intervals.

### Protein purification and GST affinity isolation experiments
GST-tagged proteins were expressed in *Escherichia coli* BL21 (DE3). GST fusion proteins were purified on glutathione–Sepharose 4 Fast Flow beads (GE Healthcare, 17513201) followed by washing with NET-N buffer [100 mM NaCl, 1 mM EDTA, 0.5% Nonidet P-40 (Sigma-Aldrich, 74385), 50 mM Tris-HCl pH 8] supplemented with cOmplete Mini EDTA-free protease inhibitor mixture tablets (Roche Applied Science, 11836170001). GST-tagged proteins were eluted with 50 mM Tris-HCl pH 8, 200 mM NaCl, 5 mM reduced L-glutathione (Sigma-Aldrich, G425). GST affinity isolation assays were performed with $^{35}$S-labeled proteins co-transcribed and translated using the TNT Coupled Reticulocyte Lysate System (Promega, L4610) as described previously (Pankiv et al., 2007). For quantifications, gels were vacuum dried and $^{35}$S-labeled proteins detected on a Fujifilm bioimaging analyzer BAS-5000 (Fujifilm, Tokyo, Japan).

### Western blot and immunoprecipitation experiments
For western blotting experiments, cells were washed in PBS (137 mM NaCl, 2.7 mM KCl, 4.3 mM Na$_2$HPO$_4$, 1.47 mM KH$_2$PO$_4$, pH 7.4). followed by lysis directly in SDS-PAGE loading buffer (2% SDS, 10% glycerol 50 mM Tris-HCl, pH 6.8) and boiled for 10 min. Protein concentration was measured followed by the addition of Bromophenol Blue (0.1%) and DTT (100 mM). Samples (20 µg) were run on 10–16% gradient or 10% SDS-polyacrylamide gels and blotted on Hybond nitrocellulose membranes (GE Healthcare, 10600003) followed by Ponceau S staining. Blocking was performed in 5% nonfat dry milk in PBS with Tween 20 (0.1%). The primary antibody was diluted in PBS with Tween 20 containing 5% nonfat dry milk and incubation was performed overnight at 4°C. Secondary antibody incubation was performed at room temperature for 1 h in PBS with Tween 20 containing 5% nonfat dry milk. Membranes were washed three times before the addition of secondary antibody and development using LAS-4000 (Fujifilm, Tokyo, Japan). Images of uncropped blots from this study are shown in Fig. S7.

### Bioinformatics and statistics
The prediction of the NoLS and NES site was performed using online servers: http://www.compbio.dundee.ac.uk/www-nod/ and https://services.healthtech.dtu.dk/services/NetNES-1.1/ , respectively. Data in all figures are presented as mean±s.d. from a minimum of three independent experiments, unless otherwise specified. Statistical analysis was conducted using one-way ANOVA, followed by Tukey's multiple comparisons test in GraphPad Prism. Significance levels are indicated as follows: ns, $P$>0.05; *$P$<0.01; **$P$<0.005; ***$P$<0.001.

### Acknowledgements
We thank Nils-Anders Johannes Labba for technical assistance. We are grateful to the proteomics and imaging core facilities at UiT, Faculty of Health Sciences, for valuable assistance.

### Competing interests
The authors declare no competing or financial interests.

### Author contributions
Conceptualization: B.K.S., E.S., T.L., T.J.; Data curation: B.K.S., J.-A.B.; Formal analysis: B.K.S., E.S., J.-A.B.; Funding acquisition: T.J.; Investigation: B.K.S., E.S., H.L.O., I.O., A.Ø., H.B.B., J.-A.B.; Methodology: B.K.S., E.S., I.O., J.-A.B.; Project administration: T.J.; Resources: B.K.S., J.-A.B., T.J.; Supervision: T.L., T.J.; Validation: B.K.S., E.S., J.-A.B.; Visualization: B.K.S., T.J.; Writing – original draft: B.K.S., E.S., H.L.O., J.-A.B., T.L., T.J.; Writing – review & editing: B.K.S., H.L.O., T.J.

### Funding
This work was funded by grants from the FRIBIOMED (grant number 214448) and the TOPPFORSK (grant number 249884) programs of the Research Council of Norway (Norges Forskningsråd), and the Norwegian Cancer Society (Kreftforeningen;

grant number 71043-PR-2006-0320) to T.J. Open Access funding provided by UiT The Arctic University of Norway. Deposited in PMC for immediate release.

**Data and resource availability**
All relevant data and details of resources can be found within the article and its supplementary information.

**Peer review history**
The peer review history is available online at https://journals.biologists.com/jcs/lookup/doi/10.1242/jcs.264267.reviewer-comments.pdf

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
