## [Peer Review File · Journal of Cell Science]

A C-terminal cytoplasmic retention motif and nuclear localization signal regulates nuclear import of TP53INP2

Birendra Kumar Shrestha, Eva Sjøttem, Hallvard Lauritz Olsvik, Isaac Odonkor, Aud Øvervatn, Hanne Britt Brenne, Jack-Ansgar Bruun, Trond Lamark and Terje Johansen
DOI: 10.1242/jcs.264267

Editor: Simon Cook

Review timeline

Original submission:	3 July 2025
Editorial decision:	23 July 2025
First revision received:	24 October 2025
Accepted:	12 November 2025

Original submission

First decision letter

MS ID#: jcs.264267

MS TITLE: A C-terminal cytoplasmic retention motif and nuclear localization signal regulates nuclear import of TP53INP2/DOR

AUTHORS: Birendra Kumar Shrestha; Eva Sjøttem; Hallvard Lauritz Olsvik; Isaac Odonkor; Aud Øvervatn; Hanne Britt Brenne; Jack-Ansgar Bruun; Trond Lamark; Terje Johansen

ARTICLE TYPE: Research Article

Dear Prof Johansen,

We have now reached a decision on the above manuscript.

To see the reviewers' reports and a copy of this decision letter, please go to:

As you will see, both reviewers have provided favourable reports but raised some areas where they feel the manuscript could be improved. I hope that you will be able to carry these out because I would like to be able to accept your paper, depending on further comments from reviewers.

Reviewer 1

The paper by Shrestha et al examines the localization and putative autophagic function of the transcriptional coactivator TP53INP2. Using inducible cell lines expressing low levels of the protein and a number of domain or point mutants, a near complete itinerary of TP53INP2 is described which suggests that this normally nuclear protein translocates to the cytoplasm upon stress. Its unusually fast degradation depends primarily on the proteasome but it appears not to involve direct ubiquitination. The authors do not verify results from any of the previous papers (several exist) showing a role for TP53INP2 in autophagosome formation, in trafficking of LC3 out of the nucleus or in autophagic degradation of the protein upon starvation or mTOR inactivation.

This is an important paper which highlights how transient overexpression can lead to wrong conclusions. This is likely the explanation for the discrepancy between what the authors report here and the previous papers assigning a role for TP53INP in autophagy-all done using transient

overexpression. Although the authors consider this a result of overexpression leading to saturation of cellular control systems I think that it may also depend on the actual process of transfection, i.e. transient but massive exposure of cells to liposomes and nucleic acids activating various response pathways.

The data are very convincing although I am still not sure what happens when pathological conditions lead to TP53INP2 colocalizing with LC3 puncta. Are these genuine autophagosomal structures reflecting a level of basal autophagy? Do they contain early markers for example? If this does not lead to autophagosomal degradation - which I think it does not - when does the protein exit the autophagic structures? The only data showing that a fragment of TP53INP2 is partially degraded by autophagy (Fig 3C) are not very strong and certainly not quantitated. [In this context, the paper offers another important lesson, namely that LC3 going into punctate distribution should not be taken on its own as evidence for an autophagic response.]

I am also puzzled by the various conditions that lead to cytoplasmic localization of TP53INP2 expressed at low levels (Fig S2). What links those stress conditions in terms of the TP53INP2 effect? Presumably they are not reflecting mTOR inactivation?

Are the puncta shown for some of the domain mutants in Fig 2B all autophagy related, and is the delta 189-211 construct localising in autophagic puncta even under basal conditions?

The authors highlight throughout that their inducible expression system may more faithfully mimic endogenous expression of TP53INP2 but do we actually know the levels of the endogenous protein?

These questions, which can be addressed rapidly, would improve this paper.

Reviewer 2

This manuscript presents an extensive study uncovering the mechanisms regulating subcellular localisation of TP53INP2 (also known as DOR), a multifunctional protein previously implicated in transcriptional coactivation, ribosome biogenesis, and autophagy. Using a knockout and stable reconstitution strategy, the authors demonstrate that nuclear import of TP53INP2 is governed by a C-terminal nuclear localisation signal overlapping a nucleolar localisation signal, while its cytoplasmic retention during cellular stress is mediated by a conserved nine-amino acid cytoplasmic retention motif (CRM). Notably, the study challenges previous models by showing that TP53INP2 does not undergo active nuclear export under starvation; instead, nuclear import is inhibited, resulting in cytoplasmic accumulation. The authors also reveal that this localisation switch is conserved in TP53INP1 and that starvation-induced TP53INP2 accumulation is promoted via increased translation mediated by the m6A demethylase FTO. The experimental design is elegant, the data are of high quality, and the findings provide important insights into the interplay between stress signalling, proteostasis, and subcellular localisation, which will be of broad interest to researchers working on nuclear-cytoplasmic transport, autophagy, and cellular stress responses. In particular, the concept of CRM is novel and of significant importance for cell biology. Some suggestions as outlined below may help to improve the manuscript further during revisions.

1. Based on the K3R mutant data in Fig. 1 the authors conclude that proteasomal degradation of TP53INP2 is ubiquitin-independent. Can the authors exclude ubiquitination of TP53INP2 on alternative residues (S, T, C)? Expression of K0 ubiquitin construct could help resolving this question if of interest.

2. It remains unclear how HBSS suppresses TP53INP2 interaction with Importin in pull-down assays. The authors propose that the mechanism involves differential binding of additional proteins which interact with the CRM region in FM and HBSS. If this is the case, would it be worth testing if CRM mutant/deletion constructs affect the interaction of TP53INP2 with Importin in FM and HBSS conditions?

3. Additionally, the authors tested a number of S and T residues as potential phospho-sites, however the CRM region contains two Y residues which can also be phosphorylated (Y220 is highly conserved). There is also a relatively highly conserved C residue which could be involved in PTMs

and form disulphides. Perhaps the authors would be interested in testing if mutations of these residues affect cytoplasmic retention and interaction with Importin.

4. "Increased expression" when referring to protein should be replaced with "increased levels", to avoid confusion with gene expression.
5. It would be preferable to show individual datapoints from independent experiments on all graphs for transparency and which is becoming a convention in the literature.

First revision

Author response to reviewers' comments

Point-by-point responses to the comments made by the reviewers:

Comments from the Reviewers:

Reviewer 1: The paper by Shrestha et al examines the localization and putative autophagic function of the transcriptional coactivator TP53INP2. Using inducible cell lines expressing low levels of the protein and a number of domain or point mutants, a near complete itinerary of TP53INP2 is described which suggests that this normally nuclear protein translocates to the cytoplasm upon stress. Its unusually fast degradation depends primarily on the proteasome but it appears not to involve direct ubiquitination. The authors do not verify results from any of the previous papers (several exist) showing a role for TP53INP2 in autophagosome formation, in trafficking of LC3 out of the nucleus or in autophagic degradation of the protein upon starvation or mTOR inactivation.

This is an important paper which highlights how transient overexpression can lead to wrong conclusions. This is likely the explanation for the discrepancy between what the authors report here and the previous papers assigning a role for TP53INP in autophagy-all done using transient overexpression. Although the authors consider this a result of overexpression leading to saturation of cellular control systems I think that it may also depend on the actual process of transfection, i.e. transient but massive exposure of cells to liposomes and nucleic acids activating various response pathways.

Response: We thank the reviewer for the positive response to our manuscript. We agree with the reviewer that the transfection reagents and introduction of large amounts of plasmid DNA may trigger response pathways affecting the behavior of TP53INP2.

The data are very convincing although I am still not sure what happens when pathological conditions lead to TP53INP2 colocalizing with LC3 puncta. Are these genuine autophagosomal structures reflecting a level of basal autophagy? Do they contain early markers for example? If this does not lead to autophagosomal degradation - which I think it does not - when does the protein exit the autophagic structures? The only data showing that a fragment of TP53INP2 is partially degraded by autophagy (Fig 3C) are not very strong and certainly not quantitated. [In this context, the paper offers another important lesson, namely that LC3 going into punctate distribution should not be taken on its own as evidence for an autophagic response.]

Response: We thank the reviewer for asking these important questions. When TP53INP2 is overexpressed by transient transfection some of the protein colocalize with LC3B in the cytoplasm upon starvation and there is an accumulation of the colocalized puncta upon BafA1 treatment of the starved cells. This is clearly illustrated in Supplemental Figure S3. We have made this clear in the figure legend and also added the notion of the BafA1 induced accumulation in the main text of the Results section of the revised manuscript: "In contrast, EGFP-TP53INP2 over-expressed by transient transfection in the KO cells, formed puncta that co-localized with LC3B in the cytoplasm upon starvation and accumulated in response to BafA1 treatment (Fig. S3)." Others have also observed that overexpressed TP53INP2 colocalize with autophagy markers. For example, a relatively recent paper You et al. 2019 show that overexpressed TP53INP2 colocalized to early

autophagic membrane structures containing ULK1, ATG14, ZFYVE1/DFCP1 or WIPI2 in an LC3-dependent manner. Mauvezin et al. 2010 showed that TP53INP2 localized to early autophagosomes, but not to autolysosomes. TP53INP2 overexpression enhanced autophagosome formation (EM and GFP-LC3 positive puncta). While not being an autophagy substrate TP53INP2 is in a number of papers referred to as a positive regulator or activator of autophagy (Nowak et al. 2009, Mauvezin et al. 2010, Sala et al. 2014, Huang et al. 2015, You et al., 2019, and Ivanova et al. 2019). The mentioned papers are all in the Reference list of our paper. When the reviewer is asking “Are these genuine autophagosomal structures reflecting a level of basal autophagy?” we have not looked at this as this is not the focus of our work, but referring to the papers mentioned above this can be interpreted as an induced autophagy level caused by TP53INP2 overexpression. It is only a small fraction of TP53INP2 that colocalize with LC3 and it likely does not amount to a significant autophagic degradation. The level of TP53INP2 is regulated by proteasomal degradation, not by autophagy. As the referee pointed out “The only data showing that a fragment of TP53INP2 is partially degraded by autophagy (Fig 3C) are not very strong and certainly not quantitated.” We agree and have performed several new experiments and quantified the levels of EGFP-TP53INP2(Δ 189-211) in the western blot shown in Fig. 3C with the quantifications shown in new Figure 3D in the revised manuscript. We do not see a statistically significant increase in the protein levels of EGFP-TP53INP2(Δ 189-211) upon inhibition of lysosomal degradation. We have therefore changed the main text in the revised version now stating clearly that “Monitoring the expression level of EGFP-TP53INP2(Δ 189-211) by Western blotting did not reveal a statistically significant accumulation of the NLS-NoLS mutant upon inhibition of lysosomal degradation. Also, the NLS-NoLS mutant was profoundly stabilized by proteasome inhibition (Fig. 3C and D).” We have also changed the subheading in Results and the heading of Figure legend to Fig 3 by changing “Deletion of the TP53INP2 NLS leads to accumulation in LC3B-positive cytoplasmic puncta and some degradation via autophagy” to “Deletion of the TP53INP2 NLS leads to accumulation in LC3B-positive cytoplasmic puncta”.

I am also puzzled by the various conditions that lead to cytoplasmic localization of TP53INP2 expressed at low levels (Fig S2). What links those stress conditions in terms of the TP53INP2 effect? Presumably they are not reflecting mTOR inactivation?

Response: We thank the reviewer for this insightful question. As shown, some of the stress conditions are dependent on mTOR inactivation like amino acid starvation and mTOR kinase inhibitors, while others are not, like CCCP, thapsigargin, sodium arsenite and puromycin. A common denominator or shared output response for these stresses is a global slowdown of translation. This occurs obviously upon mTOR inhibition but also via the integrated stress response where thapsigargin will give ER stress and lead to PERK activation phosphorylating eIF2 α , arsenite can lead to HRI activation and eIF2 α phosphorylation, puromycin gives premature chain termination and provoke ribosome stalling and ribosome stress signaling while CCCP gives energetic stress that also ultimately may affect translation.

Are the puncta shown for some of the domain mutants in Fig 2B all autophagy related, and is the delta 189-211 construct localising in autophagic puncta even under basal conditions?

Response: The TP53INP2 delta 189-211 colocalizes with LC3B puncta even under basal conditions. However, there are few puncta that colocalize with LC3B dots. We do not know the nature of the puncta that does not colocalize with LC3B.

The authors highlight throughout that their inducible expression system may more faithfully mimic endogenous expression of TP53INP2 but do we actually know the levels of the endogenous protein?

Response: The short answer is no. We have tested several (3-4) commercially available antibodies, but none recognized EGFP-TP53INP2 in our cells. So, it is difficult to provide a definitive answer here. Publicly available data indicate that mRNA expression levels are quite low.

Reviewer 2: This manuscript presents an extensive study uncovering the mechanisms regulating subcellular localisation of TP53INP2 (also known as DOR), a multifunctional protein previously implicated in transcriptional coactivation, ribosome biogenesis, and autophagy. Using a knockout and stable reconstitution strategy, the authors demonstrate that nuclear import of TP53INP2 is

governed by a C-terminal nuclear localisation signal overlapping a nucleolar localisation signal, while its cytoplasmic retention during cellular stress is mediated by a conserved nine-amino acid cytoplasmic retention motif (CRM). Notably, the study challenges previous models by showing that TP53INP2 does not undergo active nuclear export under starvation; instead, nuclear import is inhibited, resulting in cytoplasmic accumulation. The authors also reveal that this localisation switch is conserved in TP53INP1 and that starvation-induced TP53INP2 accumulation is promoted via increased translation mediated by the m6A demethylase FTO. The experimental design is elegant, the data are of high quality, and the findings provide important insights into the interplay between stress signalling, proteostasis, and subcellular localisation, which will be of broad interest to researchers working on nuclear-cytoplasmic transport, autophagy, and cellular stress responses. In particular, the concept of CRM is novel and of significant importance for cell biology. Some suggestions as outlined below may help to improve the manuscript further during revisions.

Response: We thank the reviewer for the very positive comments.

1. Based on the K3R mutant data in Fig. 1 the authors conclude that proteasomal degradation of TP53INP2 is ubiquitin-independent. Can the authors exclude ubiquitination of TP53INP2 on alternative residues (S, T, C)? Expression of K0 ubiquitin construct could help resolving this question if of interest.

Response: We cannot exclude that TP53INP2 can be ubiquitinated on alternative residues (S, T or C). However, if this occurs it must be a minor event as we do not see any upshifted band in Western blots. None of all the different deletion and point mutants of S and T residues we studied in Supplemental Figure S5 showed any change of the subcellular distribution in full medium relative to starvation medium as analyzed by confocal microscopy. This suggests no major impact of any of the S or T mutations on the degradation of the protein.

2. It remains unclear how HBSS suppresses TP53INP2 interaction with Importin in pull-down assays. The authors propose that the mechanism involves differential binding of additional proteins which interact with the CRM region in FM and HBSS. If this is the case, would it be worth testing if CRM mutant/deletion constructs affect the interaction of TP53INP2 with Importin in FM and HBSS conditions?

The deletion of CRM (EGFP-TP53INP2 1-211) leads to degradation of TP53INP2 under starvation as the construct is imported into the nucleus and degraded there by the proteasome (see Figure 6D). It will not be an alternative to test the importin interaction with recombinant TP53INP2 1-211 since the protein(s) involved in retaining TP53INP2 in the cytoplasm upon starvation are not included.

3. Additionally, the authors tested a number of S and T residues as potential phospho-sites, however the CRM region contains two Y residues which can also be phosphorylated (Y220 is highly conserved). There is also a relatively highly conserved C residue which could be involved in PTMs and form disulphides. Perhaps the authors would be interested in testing if mutations of these residues affect cytoplasmic retention and interaction with Importin.

Response: By including more species in the multialignment of the CRM from TP53INP2 proteins we see that the C residue is not very conserved. We have mutated the conserved C-terminal Y220 to F and R216 to K and also to N. These mutants (Y220F, R216K and R216N) behaved like WT when expressed in HeLa cells and studied by confocal microscopy for subcellular localization. We have not included any of these data since these mutants did not affect cytoplasmic retention.

4. "Increased expression" when referring to protein should be replaced with "increased levels", to avoid confusion with gene expression.

Response: We are grateful to the reviewer for pointing this out and have corrected to "increased levels" in the revised version.

5. It would be preferable to show individual datapoints from independent experiments on all graphs for transparency and which is becoming a convention in the literature.

Response: We thank the reviewer for pointing this out and have replaced all bar graphs in the original manuscript, in both main figures and supplementary figures, with new graphs where the individual data points are shown.

Second decision letter

MS ID#: jcs.264267R1

MS Title: A C-terminal cytoplasmic retention motif and nuclear localization signal regulates nuclear import of TP53INP2/DOR

Authors: Birendra Kumar Shrestha; Eva Sjøttem; Hallvard Lauritz Olsvik; Isaac Odonkor; Aud Øvervatn; Hanne Britt Brenne; Jack-Ansgar Bruun; Trond Lamark; Terje Johansen
Article Type: Research Article

Dear Prof Johansen,

I am happy to tell you that your manuscript has been accepted for publication in Journal of Cell Science, pending standard publication integrity checks.